# High-dimensional robust regression under heavy-tailed data: Asymptotics and Universality

## Abstract

We investigate the high-dimensional properties of robust regression estimators in the presence of heavy-tailed contamination of both the covariates and response functions. In particular, we provide a sharp asymptotic characterisation of M-estimators trained on a family of elliptical covariate and noise data distributions including cases where second and higher moments do not exist. We show that, despite being consistent, the Huber loss with optimally tuned location parameter $\delta$ is suboptimal in the high-dimensional regime in the presence of heavy-tailed noise, highlighting the necessity of further regularisation to achieve optimal performance. This result also uncovers the existence of a curious transition in $\delta$ as a function of the sample complexity and contamination. Moreover, we derive the decay rates for the excess risk of ridge regression. We show that, while it is both optimal and universal for noise distributions with finite second moment, its decay rate can be considerably faster when the covariates' second moment does not exist. Finally, we show that our formulas readily generalise to a richer family of models and data distributions, such as generalised linear estimation with arbitrary convex regularisation trained on mixture models.

## 1 Introduction

Consider the classical statistical problem of estimating a vector $\boldsymbol{\beta}_\star \in \mathbb{R}^d$ from $n$ i.i.d. pairs of observations $\mathcal{D} \coloneqq \{(\boldsymbol{x}_i, y_i) \in \mathbb{R}^{d+1} : i \in [n]\}$ from a linear model:

$$y_i = \boldsymbol{\beta}_\star^\mathsf{T} \boldsymbol{x}_i + \eta_i, \tag{1a}$$

where $\boldsymbol{x}_i \in \mathbb{R}^d$ are the covariates and $\eta_i \in \mathbb{R}$ is the label noise, which we assume to be a random quantity with zero mean and independent of the covariates. This manuscript is concerned with the characterisation of the following class of (regularised) M-estimators:

$$\hat{\boldsymbol{\beta}}_\lambda \coloneqq \arg\min_{\boldsymbol{\beta} \in \mathbb{R}^d} \sum_{i=1}^n \rho(y_i - \boldsymbol{\beta}^\mathsf{T} \boldsymbol{x}_i) + \frac{\lambda}{2}\|\boldsymbol{\beta}\|_2^2, \qquad \lambda \in \mathbb{R}_+ \coloneqq [0, +\infty), \tag{1b}$$

with $\rho \colon \mathbb{R} \to \mathbb{R}_+$ a convex objective function. A popular particular example is least-squares regression, where $\rho(t) = t^2/2$ and $\lambda = 0$: in this case, $\hat{\boldsymbol{\beta}}_\lambda$ is the maximum likelihood estimator for $\boldsymbol{\beta}_\star$ if $\eta_i \sim \mathcal{N}(0, 1)$. It is well-known, however, that the least-squares estimator suffers in the presence of outliers in the data (Huber, 1973). Indeed, the fact that the gradient of the loss $\rho'(t) = t$ is unbounded implies that an outlier can have an arbitrary influence over the solution of Eq. 1b. Tailoring the objective $\rho$ to be insensitive (i.e., *robust*) to outliers in the training data $\mathcal{D}$ is a classical statistical problem (Huber, 2004; Hampel et al., 2011; Rousseeuw and Leroy, 2005; Maronna et al., 2019). In his seminal work, Huber (1964) has shown that judiciously trimming the squared loss

$$\rho_\delta(t) = \begin{cases} t^2/2 & \text{if } |t| < \delta \\ \delta|t| - \delta^2/2 & \text{otherwise} \end{cases} \tag{2}$$

| | $\varrho(\sigma)$ | $p(x) = \mathbb{E}_\sigma[\mathcal{N}(x; 0, \sigma^2)]$ | $k$th moment exists if |
|---|---|---|---|
| Inverse-Gamma | $\frac{2b^a \exp(-b/\sigma^2)}{\Gamma(a)\sigma^{2a+1}}$ | $\frac{(2b)^a \Gamma(a+1/2)}{\sqrt{\pi}\Gamma(a)(2b+x^2)^{a+1/2}}$ | $k < 2a$ |
| for $a = b = 1/2$ | | $\frac{1}{\pi}\frac{1}{1+x^2}$ (Cauchy) | none |
| for $2a = 2b = n$ | | $\frac{\Gamma(\frac{n+1}{2})}{\sqrt{\pi n}\Gamma(n/2)}\left(1+\frac{x^2}{n}\right)^{-\frac{n+1}{2}}$ (Student-t) | $k < n$ |
| for $a = 1 + b \to +\infty$ | | $\frac{1}{\sqrt{2\pi}}\,\mathrm{e}^{-\frac{x^2}{2}}$ (Gaussian) | all |
| Pareto | $\frac{2a\theta(\sigma-1)}{\sigma^{2a+1}}$ | $\frac{a2^a\gamma(a+1/2,1/2x^2)}{\sqrt{\pi}|x|^{2a+1}}$ | $k < 2a$ |
| for $a \to +\infty$ | | $\frac{1}{\sqrt{2\pi}}\,\mathrm{e}^{-\frac{x^2}{2}}$ (Gaussian) | all |

Table 1: Concrete examples of distributions $p(x) = \mathbb{E}_\sigma[\mathcal{N}(x; 0, \sigma^2)]$ for different densities $\varrho$ of $\sigma$. Here $\gamma(a, x)$ is the lower incomplete Gamma function. Note that relevant distributions, such as the Cauchy and the Student's $t$-distribution, appear as special cases. The last column shows the values of the parameters of $\varrho$ for which the $k$th moment of the distribution is finite.

for $\delta \geq 0$ provides a convenient solution to this problem while preserving the convexity of the task in Eq. 1b. Indeed, besides enjoying standard statistical guarantees in the classical limit of $n \to \infty$, such as consistency and asymptotic normality (Huber, 1973; van der Vaart, 2000), the so-called *Huber loss* has been shown to be optimal in different regards. For instance, Huber (1964) has shown it achieves minimax asymptotic variance under symmetric contamination of the normal distribution. It also has the smallest asymptotic variance among losses with bounded sensitivity to outliers, as it can formalised by Hampel's influence function (Hampel, 1974). However, while these guarantees are fit for a classical statistical regime where data is abundant ($n \gg d$), they fall short in modern tasks where the number of features can be comparable to the quantity of data ($n \approx d$). Investigating the properties of estimators in the proportional *high-dimensional regime* where $n, d \to \infty$ at fixed sample complexity $\alpha = n/d$ has been a major endeavor in the statistical literature in the past decade, where it has been shown that standard guarantees for the maximum likelihood estimator, such as unbiasedness (Javanmard and Montanari, 2018; Sur and Candès, 2019; 2020; Bellec et al., 2022; Zhao et al., 2022) and calibration (Bai et al., 2021; Clarté et al., 2023a;b) break down in this regime. However, the majority of these works have focused in the case of sub-Gaussian features and bounded noise variance. Our goal in this manuscript is to go beyond this assumption by providing a high-dimensional characterisation of M-estimators for a family of heavy-tailed distributions for both the covariates and noise distributions.

**Heavy-tailed data** — In the following, we consider a family of covariate distributions parametrised as:

$$\boldsymbol{x}_i = \sigma_i \boldsymbol{z}_i, \qquad \boldsymbol{z}_i \sim \mathcal{N}(\boldsymbol{0}, 1/d\boldsymbol{I}_d) \qquad \sigma_i \sim \varrho, \tag{3}$$

where for each $i \in [n]$, $\sigma_i \in \mathbb{R}_+^* := (0, +\infty)$ is an independent random variable with probability density $\varrho$ supported on the positive real line. This class of covariates appeared under different contexts in physics (Beck, 2003), statistics (El Karoui, 2018; Adomaityte et al., 2023) and signal processing (Wainwright and Simoncelli, 1999) where it has been shown that by judiciously choosing $\varrho$ yields a large family of *power-law tailed distributions* for $\boldsymbol{x}$, see Table 1 for concrete examples. In particular, we are interested in investigating the impact of heavy-tail contamination of both the covariates and responses, and the sensitivity of M-estimators of the type in Eq. 1b to them. More concretely, in the following we consider the *Huber $\epsilon$-contamination model* for the covariates by assuming

$$p(\boldsymbol{x}) = \mathbb{E}[\mathcal{N}(\boldsymbol{x}; \boldsymbol{0}, \sigma^2/d\boldsymbol{I}_d)] \quad \text{with} \quad \sigma \sim \varrho = (1 - \epsilon_{\mathrm{c}})\delta_{\sigma,1} + \epsilon_{\mathrm{c}}\varrho_0, \quad \epsilon_{\mathrm{c}} \in [0, 1], \tag{4}$$

where $\varrho_0$ is a density over $\mathbb{R}_+^*$. The distribution $p(\boldsymbol{x})$ belongs therefore to the family defined by Eq. 3: the quantity $\epsilon_{\mathrm{c}}$ measures, therefore, the *contamination* of a pure Gaussian covariate distribution (recovered for $\epsilon_{\mathrm{c}} = 0$) with a possibly heavy-tailed law. Moreover, our results can be extended to the case of *mixtures*, i.e., the case of the covariates grouped in $K$ different clouds. This generalisation is introduced and briefly discussed in Section 4.

**Main contributions** — The **key contributions** in this manuscript are:

- We provide an asymptotic characterisation of the statistics of the M-estimator $\hat{\boldsymbol{\beta}}_\lambda$ defined by Eqs. 1 with heavy-tailed covariates and general label noise distributions in the proportional high-dimensional regime.
- Further, we provide a similar high-dimensional characterisation for the performance of the optimal Bayesian estimator in this problem. These two results follow from an extension of the replica method for generalised linear estimation to covariate distributions in the family of Eq. 3, and hold for a broader mixture data model which we discuss in Section 4.
- Leveraging the characterisation above, we investigate the impact of heavy-tailed covariate and response contamination on the performance of M-estimators, including, for the first time for this model, analytic control of infinite-variance covariates. In particular, our result highlights the necessity of regularising the Huber loss in the high-dimensional regime to achieve optimal performance, as we show that it can be suboptimal in the presence of heavy-tailed response contamination even when the location parameter is optimally tuned, at odds with the optimality results of Huber (1964) and Hampel (1974) in the classical regime where $n \gg d$.
- We show that, despite the strong impact of heavy-tailed contamination in the high-dimensional regime where $n/d = \Theta(1)$, the error decay rates $\|\hat{\boldsymbol{\beta}} - \boldsymbol{\beta}_\star\|_2 = \Theta(n^{-1/2})$ of optimally regularised Huber and least-squares regression are optimal provided the second moment of both the covariates and the noise are bounded. In contrast, we show that when the covariates' second moment doesn't exist (but under the assumption of a finite second moment for the noise), the rates explicitly depend on the tail behavior of the covariates' distribution.

**Related works** — Robust regression is a classical topic in statistics, with several books dedicated to the subject (Huber, 2004; Hampel et al., 2011; Rousseeuw and Leroy, 2005; Maronna et al., 2019). In contrast to the classical regime, literature on robust regression in the high-dimensional regime remains relatively scarce. Early works in this direction provided a characterisation of M-estimators in the proportional limit of $n, d \to \infty$ with fixed $\alpha = n/d$ for Gaussian (El Karoui et al., 2013; El Karoui, 2013; Donoho and Montanari, 2016) and sub-Gaussian (El Karoui, 2013) designs with bounded noise variance. In particular, it was shown that in this regime the maximum likelihood estimator is not necessarily the optimal choice of objective $\rho$ (El Karoui, 2013). De-biasing and confidence intervals for high-dimensional M-estimation on Gaussian covariates were discussed by Bellec et al. (2022); Bellec (2023). A novel de-biasing framework for covariates with possibly heavy-tailed and asymmetric distributions was recently introduced by Li and Sur (2023). The family of distributions defined in Eq. 3 has previously appeared under different names and literatures, such as *superstatistics* in the context of statistical physics (Beck, 2003), *elliptical distributions* in the context of statistics (Couillet et al., 2015; El Karoui, 2018; Adomaityte et al., 2023) and *Gaussian scale mixture* in the context of signal processing (Wainwright and Simoncelli, 1999). It has been studied in the context of high-dimensional robust covariance estimation by Couillet et al. (2015; 2016). El Karoui (2018) considered (unregularised) robust regression for elliptically distributed covariates under boundedness conditions on the moments. Our work differs in many directions: (a) we relax the assumption of bounded moments and extend the analysis to the more general case of a *mixture* of elliptical distributions; (b) we derive the asymptotic performance of the *Bayes optimal estimator* for this family; (c) we consider a generic convex penalty; (d) we investigate the impact of Huber $\epsilon$-contamination. More recently, Vilucchio et al. (2023) studied the model Eq. 1a under a Gaussian design and a double Gaussian noise model for outliers in the proportional high-dimensional regime, showing that estimators of the type in Eq. 1b can fail to be consistent. Beyond (sub-)Gaussian designs, high-dimensional upper bounds on the regressor mean-squared error were obtained under different settings (Hsu and Sabato, 2016; Lugosi and Mendelson, 2019), including heavy-tailed noise (Sun et al., 2020) and heavy-tailed covariate contamination (Sasai, 2022; Pensia et al., 2021). Heavy-tailed covariates have also been studied in the context of kernel ridge regression in (Tomasini et al., 2022).

## 2 High-dimensional asymptotics

In this section, we discuss our two main theoretical results: the high-dimensional asymptotic characterisation of the M-estimator defined in Eq. 1b and the corresponding Bayes-optimal error. In fact, the results in this section hold under the following slightly more general assumptions.

**Assumption 2.1** (Data). *The covariates $\boldsymbol{x}_i \in \mathbb{R}^d$, $i \in [n]$ are independently drawn from the family of "superstatistical" distributions $\boldsymbol{x}_i \sim p(\boldsymbol{x}) := \mathbb{E}[\mathcal{N}(\boldsymbol{x}; \boldsymbol{0}, \sigma^2/d\boldsymbol{I}_d)]$, where the expectation is over $\sigma$ with generic distribution density $\varrho$ supported on the positive real line $\mathbb{R}_+^* := (0, +\infty)$. For each $i \in [n]$, the corresponding response $y_i \in \mathcal{Y}$ is drawn from a conditional law $P_0$ on $\mathcal{Y}$:*

$$y_i \sim P_0(\cdot | \boldsymbol{\beta}_\star^\mathsf{T} \boldsymbol{x}_i), \tag{5}$$

*with target weights $\boldsymbol{\beta}_\star \in \mathbb{R}^d$ having finite normalised norm $\beta_\star^2 := \lim_{d\to\infty} {}^1/d \|\boldsymbol{\beta}_\star\|_2^2$.*

Note that the model in Eq. 1a corresponds to the choice $P_0(y|\tau) = p_\eta(\eta - \tau)$.

**Assumption 2.2** (Predictor). *We consider the hypothesis class of generalised linear predictors $\mathcal{H} = \{f_\beta(\boldsymbol{x}) := f(\boldsymbol{\beta}^\mathsf{T}\boldsymbol{x}), \boldsymbol{\beta} \in \mathbb{R}^d\}$, where $f: \mathbb{R} \to \mathcal{Y}$ is a generic activation function, and the weights $\boldsymbol{\beta} \in \mathbb{R}^d$ are obtained by minimising the following empirical risk:*

$$\hat{\boldsymbol{\beta}}_\lambda := \underset{\boldsymbol{\beta} \in \mathbb{R}^d}{\arg\min} \ \sum_{i=1}^n \rho\left(y_i - \boldsymbol{\beta}^\mathsf{T}\boldsymbol{x}_i\right) + \frac{\lambda}{2}\|\boldsymbol{\beta}\|_2^2, \qquad \lambda \in \mathbb{R}_+, \tag{6}$$

*for a convex objective function $\rho: \mathbb{R} \to \mathbb{R}_+$.*

**Assumption 2.3** (Proportional regime). *We consider the proportional high-dimensional regime where both $n, d \to \infty$ at a fixed ratio $\alpha := {}^n/d$, known as the sample complexity.*

In particular, note that Assumption 2.1 covers the additive noise model in Eq. 1a, and that the standard robust regression setting is given by taking $f(\boldsymbol{\beta}^\mathsf{T}\boldsymbol{x}) = \boldsymbol{\beta}^\mathsf{T}\boldsymbol{x}$. In Appendix A and Appendix B we derive the following result.

**Result 2.4** (High-dimensional asymptotics). *Let $\varphi: \mathcal{Y} \times \mathbb{R} \to \mathbb{R}$ denote a test function, and define the following generalisation and training statistics:*

$$\mathcal{E}_g(\hat{\boldsymbol{\beta}}) = \mathbb{E}_{(y,\boldsymbol{x})}\left[\varphi(y, \hat{\boldsymbol{\beta}}_\lambda^\mathsf{T}\boldsymbol{x})\right], \qquad \mathcal{E}_t(\hat{\boldsymbol{\beta}}) := \tfrac{1}{n}\sum_{i=1}^n \varphi(y_i, \hat{\boldsymbol{\beta}}_\lambda^\mathsf{T}\boldsymbol{x}_i). \tag{7}$$

*Then, under Assumptions 2.1–2.3 we have:*

$$\mathcal{E}_g(\hat{\boldsymbol{\beta}}) \xrightarrow[n,d\to\infty]{\mathrm{P}} \varepsilon_g(\alpha, \lambda, \beta_\star^2), \qquad \mathcal{E}_t(\hat{\boldsymbol{\beta}}) \xrightarrow[n,d\to\infty]{\mathrm{P}} \varepsilon_t(\alpha, \lambda, \beta_\star^2), \tag{8}$$

*explicitly given by:*

$$\begin{aligned}
\varepsilon_t(\alpha, \lambda, \beta_\star^2) &= \int_{\mathcal{Y}} \mathrm{d}y\, \mathbb{E}_{\sigma,\zeta}\left[Z_0\left(y, \tfrac{m\sigma}{\sqrt{q}}\zeta, \sigma^2\beta_\star^2 - \tfrac{\sigma^2 m^2}{q}\right)\varphi\left(y, \sigma\sqrt{q}\zeta + v\sigma^2 f\right)\right], \\
\varepsilon_g(\alpha, \lambda, \beta_\star^2) &= \int_{\mathcal{Y}} \mathrm{d}y \int \mathrm{d}\eta \int \mathrm{d}\tau\, P_0\left(y|\tau\right) \mathbb{E}_\sigma\left[\mathcal{N}\left(\binom{\tau}{\eta}; \boldsymbol{0}, \sigma^2\binom{\beta_\star^2\ \ m}{m\ \ q}\right)\right]\varphi(y, \eta).
\end{aligned} \tag{9}$$

*where $\boldsymbol{\xi} \sim \mathcal{N}(\boldsymbol{0}, \boldsymbol{I}_d)$, $\zeta \sim \mathcal{N}(0, 1)$ are independent Gaussian variables. Similarly, the mean-squared error on the estimator is given by:*

$$\varepsilon_{\mathrm{est}}(\alpha, \lambda, \beta_\star^2) := \lim_{n,d\to+\infty} \frac{1}{d}\mathbb{E}_\mathcal{D}\left[\|\hat{\boldsymbol{\beta}}_\lambda - \boldsymbol{\beta}_\star\|_2^2\right] = \beta_\star^2 - 2m + q \xrightarrow{\alpha\to+\infty} \beta_\star^{-2}\lim_{\alpha\to+\infty}(m - \beta_\star^2)^2. \tag{10}$$

*Moreover, the angle between the estimator $\hat{\boldsymbol{\beta}}_\lambda$ and $\boldsymbol{\beta}_\star$ is given by*

$$\lim_{n,d\to+\infty} \mathbb{E}_\mathcal{D}[\mathrm{angle}(\hat{\boldsymbol{\beta}}_\lambda, \boldsymbol{\beta}_\star)] = \tfrac{1}{\pi}\arccos\left(\tfrac{m}{\beta_\star\sqrt{q}}\right) \xrightarrow{\alpha\to+\infty} 0. \tag{11}$$

*In the expressions above we have introduced the order parameters $v$, $q$, and $m$. These quantities are found by solving the following set of self-consistent fixed-point equations:*

$$\begin{aligned}
q &= \tfrac{\hat{m}^2\beta_\star^2 + \hat{q}}{(\lambda + \hat{v})^2} & \hat{q} &= \alpha \int_{\mathcal{Y}} \mathrm{d}y\, \mathbb{E}_{\sigma,\zeta}\left[\sigma^2 Z_0\left(y, \tfrac{m\sigma}{\sqrt{q}}\zeta, \sigma^2\beta_\star^2 - \tfrac{\sigma^2 m^2}{q}\right)f^2\right], \\
m &= \tfrac{\beta_\star^2\hat{m}}{\lambda + \hat{v}} & \hat{v} &= -\alpha \int_{\mathcal{Y}} \mathrm{d}y\, \mathbb{E}_{\sigma,\zeta}\left[\sigma^2 Z_0\left(y, \tfrac{m\sigma}{\sqrt{q}}\zeta, \sigma^2\beta_\star^2 - \tfrac{\sigma^2 m^2}{q}\right)\partial_\omega f\right], \\
v &= \tfrac{1}{\lambda + \hat{v}} & \hat{m} &= \alpha \int_{\mathcal{Y}} \mathrm{d}y\, \mathbb{E}_{\sigma,\zeta}\left[\sigma^2 \partial_\mu Z_0\left(y, \tfrac{m\sigma}{\sqrt{q}}\zeta, \sigma^2\beta_\star^2 - \tfrac{\sigma^2 m^2}{q}\right)f\right],
\end{aligned} \tag{12a}$$

*where the auxiliary function $Z_0$ and proximal operator $f$ are defined by:*

$$Z_0(y, \mu, v) := \mathbb{E}_z[P_0(y|\mu + \sqrt{v}z)],$$
$$f := \arg\min_u \left[ \frac{\sigma^2 v u^2}{2} + \rho\left(y - \omega - \sigma^2 v u\right) \right] \in \mathbb{R}, \quad \omega = \sigma\sqrt{q}\zeta. \tag{12b}$$

Result 2.4 reduces the high-dimensional optimisation problem defined by Eq. 2.2 to a self-consistent equation on the parameters $\hat{q}, q, \hat{m}, m, \hat{v}, v \in \mathbb{R}$. Despite being cumbersome, this low-dimensional problem can be efficiently solved numerically. Note that our result for $\varepsilon_{\text{est}}$ implies that the estimator $\hat{\boldsymbol{\beta}}_\lambda$ is consistent if and only if $\lim_{\alpha \to +\infty} m = \beta_\star^2$. An interesting particular case, in which the convergence rates to zero of $\varepsilon_{\text{est}}$ can be derived explicitly, is the case of square loss, $\rho(t) = \frac{1}{2}t^2$. In Appendix B.1 it is shown the following.

**Result 2.5** (Square loss rates). *Consider the linear model in Eq. 1a with covariates as in Eq. 3. Assume that $\mathbb{E}[\eta_i] = 0$, $\hat{\sigma}_0^2 := \mathbb{E}[\eta_i^2] \in \mathbb{R}_+^*$. Let us also assume that in Eq. 3 we have $\varrho(\sigma) \sim \frac{1}{\sigma^{2a+1}}$ for $\sigma \gg 1$ with $a > 0$. Then, the ridge estimator $\hat{\boldsymbol{\beta}}_\lambda$ minimising Eq. 1b with $\rho(t) = \frac{1}{2}t^2$ is consistent and*

$$\varepsilon_{\text{est}} \underset{\alpha \gg 1}{=} \begin{cases} \frac{\hat{\sigma}_0^2}{\sigma_0^2 \alpha} + o\left(\frac{1}{\alpha}\right) & \text{if } a > 1, \quad \text{with } \sigma_0^2 := \mathbb{E}[\sigma^2] = \lim_{x \to +\infty} x(1 - x S_{\sigma^2}(x)), \\ \frac{\hat{\sigma}_0^2}{\tilde{\sigma}_0^2 \alpha \ln \alpha} + o\left(\frac{1}{\alpha \ln \alpha}\right) & \text{if } a = 1, \quad \text{with } \tilde{\sigma}_0^2 = \lim_{x \to +\infty} \frac{x}{\ln x}(1 - x S_{\sigma^2}(x)), \\ \frac{\hat{\sigma}_0^2}{(\tilde{\sigma}_0^2 \alpha)^{1/a}} + o\left(\frac{1}{\alpha^{1/a}}\right) & \text{if } a \in (0, 1), \quad \text{with } \tilde{\sigma}_0^2 = \lim_{x \to +\infty} x^a(1 - x S_{\sigma^2}(x)), \end{cases} \tag{13}$$

*where $S_{\sigma^2}(x) := \mathbb{E}\left[\frac{1}{x + \sigma^2}\right]$ is the Stieltjes transform of the measure of $\sigma^2$. Finally, the asymptotic test and training error will depend on the noise distribution through its second moment only.*

By definition, the mean-squared error $\varepsilon_{\text{est}}$ of the M-estimator in Eq. 10 is lower-bounded by the minimum mean-squared error achieved by the Bayes-optimal estimator $\hat{\boldsymbol{\beta}}$, which is given by the expectation over the Bayesian posterior $\hat{\boldsymbol{\beta}} = \mathbb{E}[\boldsymbol{\beta}|\mathcal{D}]$. Our second main contribution in this manuscript, derived in Appendix A.2, is to provide a sharp asymptotic formula for the minimum mean-squared error under the assumption that $\boldsymbol{\beta}_\star \sim \mathcal{N}(\boldsymbol{x}; \boldsymbol{0}, \beta_\star^2 \boldsymbol{I}_d)$.

**Result 2.6** (Bayes optimal error). *Consider the minimum mean-squared error for the data model defined by 2.1 with $\boldsymbol{\beta}_\star \sim \mathcal{N}(\boldsymbol{x}; \boldsymbol{0}, \beta_\star^2 \boldsymbol{I}_d)$:*

$$\mathcal{E}_{\text{BO}} := \min_{\boldsymbol{\beta} \in \mathbb{R}^d} \frac{1}{d}\mathbb{E}[\|\boldsymbol{\beta}_\star - \boldsymbol{\beta}\|_2^2] = \frac{1}{d}\mathbb{E}[\|\boldsymbol{\beta}_\star - \hat{\boldsymbol{\beta}}\|_2^2]. \tag{14}$$

*Then, in the high-dimensional limit defined by Assumption 2.3:*

$$\mathcal{E}_{\text{BO}} \xrightarrow[n,d \to \infty]{\text{P}} \varepsilon_{\text{BO}} = \beta_\star^2 - \mathsf{q}, \tag{15}$$

*where $\mathsf{q}$ is given by the solution of the following self-consistent equations:*

$$\hat{\mathsf{q}} = \alpha \int_{\mathcal{Y}} \mathrm{d}y \, \mathbb{E}_{\sigma, \zeta}\left[\sigma^2 Z_0(y, \mu, V)\left(\partial_\mu \ln Z_0(y, \mu, V)\right)^2 \Big|_{\substack{\mu = \sigma\sqrt{\mathsf{q}}\zeta \\ V = \sigma^2(\beta_\star^2 - \mathsf{q})}}\right], \quad \mathsf{q} = \frac{\beta_\star^4 \hat{\mathsf{q}}}{1 + \beta_\star^2 \hat{\mathsf{q}}} \tag{16}$$

*with $\zeta \sim \mathcal{N}(0, 1)$ and $Z_0$ given in Eq. 12b.*

Similarly to Result 2.4, Result 2.6 reduces the problem of estimating the Bayes optimal error from the computationally intractable evaluation of the high-dimensional posterior marginals to a simple two-dimensional set of self-consistent equations on the variables $\hat{\mathsf{q}}, \mathsf{q} \in \mathbb{R}$.

A detailed derivation of both Results 2.4 & 2.6 is provided in Appendix A using the replica method. Despite being non-rigorous, Results 2.4 & 2.6 provide a natural extension of rigorous results in the established literature of high-dimensional asymptotics for generalised linear estimation on Gaussian covariates to the elliptical family defined in Assumption 2.1 (El Karoui, 2013; Donoho and Montanari, 2016; Thrampoulidis et al., 2018; Barbier et al., 2019; Loureiro et al., 2022). Indeed, by taking $\varrho(\sigma) = \delta_{\sigma, \bar{\sigma}}$ for some $\bar{\sigma} \in \mathbb{R}_+^*$, Result 2.4 reduces to the rigorous formulas proven by Thrampoulidis et al. (2018), and Result 2.6 reduces to the rigorous formulas proven by Barbier et al. (2019). Nevertheless, as we will see in the next section, they produce an excellent agreement with moderately finite-size simulations, across different choices of penalty $\rho$ and covariate distribution, including cases in which the variance is infinite, see Fig. 2 for instance.

## 3 DISCUSSION

In this section, we investigate the consequences of Results 2.4 – 2.6 in the context of robust regression with heavy-tail contamination as in Eq. 4 within the model in Eq. 1a. Our theoretical results are given for *any* noise distribution $p_\eta$, but, to exemplify our findings, we will focus on the case of heavy-tailed contamination of standard Gaussian noise, in a form similar to Eq. 4:

$$p_\eta(\eta) = \mathbb{E}[\mathcal{N}(\eta; 0, \sigma^2)], \quad \text{with} \quad \sigma \sim \hat{\varrho} = (1 - \epsilon_n)\delta_{\sigma,1} + \epsilon_n \hat{\varrho}_0, \quad \epsilon_n \in [0, 1]. \quad (17)$$

As in the case of covariates, $\epsilon_n$ measures the level of contamination of the purely Gaussian noise. For concreteness, in most of the plots, we take $\boldsymbol{\beta}_\star \sim \mathcal{N}(0, \beta_\star^2 \boldsymbol{I}_d)$ and use for $\varrho_0$ and $\hat{\varrho}_0$ one of the distributions in Table 1, which provide convenient parametric families of distributions where the existence of moments is easily tunable.

### 3.1 LABEL CONTAMINATION

We start by discussing the impact of *response contamination* in high-dimensional robust regression. Since the Gaussian assumption for the noise label is widespread in statistics, we will focus our discussion on an $\epsilon$-contamination of the standard normal distribution using, for $p_\eta$, an expression as in Eq. 17, by adopting a proper $\hat{\varrho}_0$ that generates fat tails. We recall the reader that when the loss function is the negative log-likelihood of the noise $-\log p_\eta$, strong guarantees hold for the M-estimator in the classical limit $n \gg d$. In particular, the estimator is unbiased and asymptotically normal, with estimation error (defining the variance) $\|\boldsymbol{\beta}_\star - \hat{\boldsymbol{\beta}}_\lambda\|_2^2 = \Theta(n^{-1})$ (van der Vaart, 2000). Since the square loss corresponds to the negative log-likelihood of standard Gaussian noise, in the absence of contamination $\epsilon_n = 0$ the square loss attains an optimal rate as $n \to \infty$. We discuss the

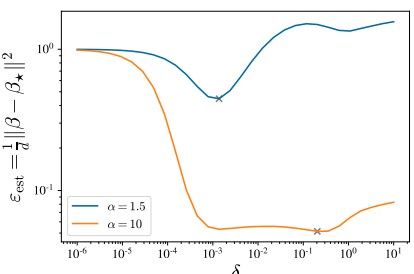

Figure 1: Value of $\varepsilon_{\text{est}}$ obtained using a regularised Huber at given $\lambda = 10^{-3}$ as a function of $\delta$ for different values of $\alpha$. Here the contamination level is $\epsilon_n = 0.5$.

insights provided by our exact high-dimensional characterisation in high dimensions, i.e., when $n = \Theta(d)$, assuming contaminated labels as in Eq. 17. On the other hand, we assume for now purely Gaussian covariates, i.e., $\epsilon_c = 0$ in Eq. 4.

**The importance of regularisation** — Fig. 2 illustrates two distinct scenarios, i.e., contamination of the labels with an infinite-variance noise (top), so that $\mathbb{E}[\eta^2] = +\infty$, and contamination with a finite-variance noise (bottom), so that $\mathbb{E}[\eta^2] = 1$. While in the former case, the performance degrades as a function of the contamination, counter-intuitively, it improves when the contamination variance is bounded. As expected, in the absence of contamination $\epsilon_n = 0$, optimally regularised ridge regression achieves the minimum mean-squared error for this model. Indeed, in this case, the square loss is not only the maximum likelihood estimator but also coincides with the Bayes-optimal estimator, since from a Bayesian perspective the $\ell_2$ penalty corresponds to the optimal Bayesian prior when optimally tuned. As contamination is introduced ($\epsilon_n > 0$), this is no longer the case, and indeed, in the $\mathbb{E}[\eta^2] < +\infty$ scenario, optimally regularised ridge regression is observed to be suboptimal. More surprising, perhaps, is that, at small values of the regularisation $\lambda$, the Huber loss in Eq. 2 with optimally chosen location parameter $\delta^\star$ is also suboptimal in the high-dimensional regime $n = \Theta(d)$. This is to be contrasted with the well-known optimality results in the classical regime $n \gg d$, which are recovered at $\alpha \to \infty$. Interestingly, at small but finite $\lambda$, the improvement of performance for the Huber loss with $\alpha$ coincides with a sharp transition of the optimal location parameter $\delta^\star$ from $O(\lambda)$ to $O(1)$ at a given value of $\alpha$: at fixed $(\lambda, \alpha)$, the value of $\varepsilon_{\text{est}}$ develops indeed two minima as a function of $\delta$, one in $\delta = O(\lambda)$ and another in $\delta = O(1)$, whose relative depth changes with $\alpha$, exhibiting an inversion in favour of the $O(1)$ minimum at large $\alpha$, see Fig. 1. For $\lambda \to 0$, the minimum in $\delta = O(\lambda)$ disappears. This phenomenology is observed for various choices of $\hat{\varrho}_0$, but

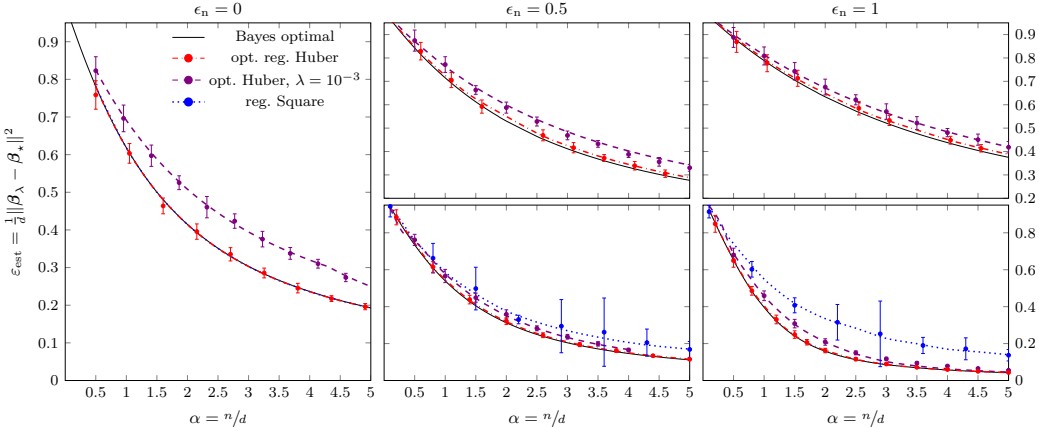

Figure 2: Response contamination of Gaussian covariates as a function of the sample complexity $\alpha = {}^n/d$ for different noise contamination levels $\epsilon_{\rm n} \in [0, 1]$ as in Eq. 17. Here we adopt $\hat{\varrho}_0(\sigma) \propto \sigma^{-2a-1} \exp\left(-b\sigma^{-2}\right)$ for the noise contamination, see Table 1. The theoretical predictions are compared with the results of numerical experiments (dots) obtained averaging over 20 instances with $d = 10^3$. (**Left**) Purely Gaussian noise ($\epsilon_{\rm n} = 0$). (**Top**). Case $b = 1$ and $a = {}^4/5 < 1$, implying $\mathbb{E}[\eta^2] = +\infty$. The performance degrades as the contamination level $\epsilon_{\rm n}$ is increased. Optimally regularised Huber (red) achieves near-optimal Bayesian performance (solid), while by fixing the regularisation, the Huber has suboptimal performances (purple). (**Bottom**). Case $a = 1 + b = 1 + {}^1/10$, corresponding to $\mathbb{E}[\eta^2] = 1$. The performance uniformly improves as the contamination $\epsilon_{\rm n}$ grows. Optimally regularised Huber (red) achieves the optimal Bayesian performance (solid), while both Huber with untuned regularisation (purple) and optimally regularised ridge (blue) are suboptimal.

it also persists in other settings, as we mention later in the text. In general, adding a $\ell_2$ regularisation and cross-validating significantly improve the performance of the Huber loss, bringing it very close to Bayes optimality at all contamination levels. Our results highlight the necessity of properly regularising robust estimators in order to achieve optimality in the high-dimensional regime. In particular, optimally regularising Huber is crucial to achieve near-Bayes-optimal performances in the case of infinite-variance label noise.

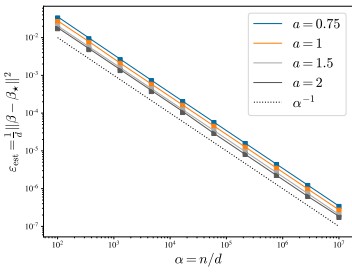

Figure 3: Estimation error $\varepsilon_{\rm est}$ as in Eq. 10 at large-$\alpha$ using regularised optimal Huber loss (solid lines). The covariates are Gaussian, whereas the label noise is obtained as in Eq. 17 with $\varrho(\sigma) \sim \sigma^{-2a-1}$, $a > 0$. The results are compared with the Bayes-optimal performance (squares). The dotted line shows a scaling of $\alpha^{-1}$.

**Convergence rates** — In Fig. 3 we consider the case of full contamination $\epsilon_{\rm n} = 1$ of the labels with both finite- and infinite-variance noise in the large-$\alpha$ regime: this corresponds to the classical limit $n \gg d$. In particular, we focus on the regularised Huber loss, whose location parameter and regularisation strength have been optimised to reach the minimum test error. A fatter tail in the noise hurts performance in terms of the estimation error $\varepsilon_{\rm est}$, but does not affect its $\alpha^{-1}$ scaling, which remains universal even when $\mathbb{E}[\eta^2] = +\infty$ and is also observed in the Bayes optimal lower bound. We will see below that this will not be the case if a contamination of the covariates is considered.

### 3.2 COVARIATE CONTAMINATION IN THE PRESENCE OF HEAVY-TAILED LABEL NOISE

We now move to discuss the impact of *covariates' contamination*, focusing on the case in which covariates are generated with a density as in Eq. 4 with $\epsilon_{\rm c} > 0$. On top of this, we assume a non-Gaussian label noise with finite variance, $\mathbb{E}[\eta^2] = 1$: note that, as discussed above, in this setting regularised ridge regression is not optimal.

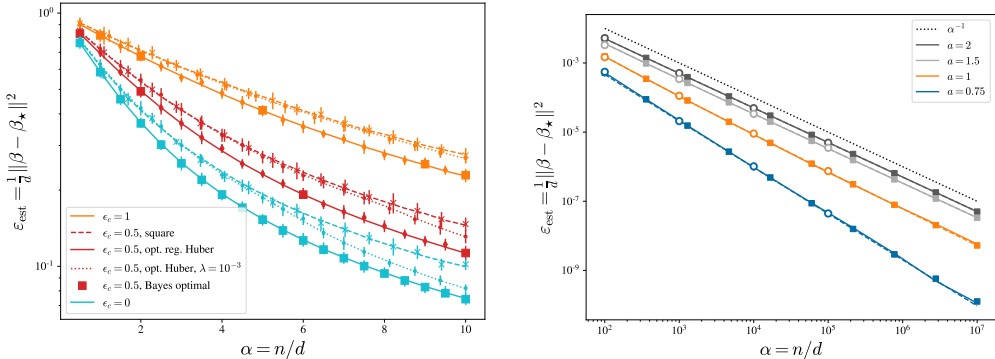

Figure 4: (**Left**) Estimation error as a function of sample complexity $\alpha$. Dots correspond to the average error of 50 numerical experiments in dimension $d = 10^3$. Covariates are contaminated as in Eq 4 using for $\varrho$ an inverse-Gamma with $b = a - 1 = {}^1/_{10}$. (**Right**) Estimation error $\varepsilon_{\text{est}}$ as in Eq. 10 at large-$\alpha$ obtained from our theory for the square loss (dashed line), regularised optimal Huber loss (solid line) and Bayes-optimal performance given by Result 2.6 (squares). The covariates' noise is Pareto-distributed to have $\varrho(\sigma) \sim \sigma^{-2a-1}$ with $a > 0$, and the label noise is Gaussian. White dots correspond to numerical experiments in dimension $d = 50$. The black dotted line shows a scaling of $\alpha^{-1}$.

Fig. 4 (left) shows that as the contamination level of the covariates $\epsilon_c$ grows, the performance is negatively affected for all metrics. The phenomenology is similar to the one observed for response contamination. While optimally regularised square loss performs the worst amongst the considered losses, an optimally $\delta$-tuned Huber loss, ridge-regularised with small, fixed strength $\lambda$, performs halfway between the square and the Bayes-optimal bound, approaching the latter as $\alpha$ grows. As observed in the previous section, the improvement in the $\delta$-tuned Huber loss occurs, for small but non-zero $\lambda$, with a sharp jump in the optimal $\delta^*$ parameter from a value $O(\lambda)$ to a value $O(1)$ as $\alpha$ increases, reflected in a kink in the curve of $\varepsilon_{\text{est}}$ as function of $\alpha$. Finally, an optimally $(\delta, \lambda)$-tuned regularised Huber loss achieves Bayes-optimal performance. The phenomenology is therefore analogous to the one described in section 3.1.

**Convergence rates** — As we show in Result 2.5, if the statistician adopts the square loss (hence choosing $\rho(t) = t^2/2$) in the study of power-law tailed covariates, the estimation error rate might depend on the tail, under the assumption that the second moment of the noise is finite. In particular, Result 2.4 implies that, in this case, the estimation error rate depends only on the covariates' second moment. As a consequence the rate $\varepsilon_{\text{est}} \sim n^{-1}$ is universal as long as the second moment of the covariates exists. Curiously, if the second moment does not exist, the estimation error decays *faster*, depending explicitly on the tail exponent of the distribution. Remarkably, this suggests that the presence of very fat tails in the covariates can actually improve the classical convergence rates of M-estimation when $n \to \infty$. Indeed, despite not having a closed form expression for the rates under different other penalties, we have numerically verified that the convergence rates reported in Result 2.4 are Bayes-optimal and are also attained optimally tuned and regularised Huber loss in different scenarios. This is illustrated in Fig. 4 (right), which clearly shows the dependence on the tail exponent of the covariate distribution when the latter has no finite second moment.

## 4 Generalised linear estimation on an elliptical mixture

We conclude our work by stating the most general form of our exact high-dimensional asymptotic result, valid for an arbitrary mixture of $K$ elliptical distributions, that we call *elliptical mixture model* (EMM). To be more precise, let us consider the following general Assumptions:

**Assumption 4.1** (EMM Data). *In the dataset $\mathcal{D} := \{(y_i, \boldsymbol{x}_i)\}_{i \in [n]}$, the covariates $\boldsymbol{x}_i \in \mathbb{R}^d$, $i \in [n]$ are independently drawn from the mixture $\boldsymbol{x}_i \sim p(\boldsymbol{x}) := \sum_{c=1}^{K} p_c \mathbb{E}_{\sigma_c}[\mathcal{N}(\boldsymbol{x}; \boldsymbol{\mu}_c, \sigma_c^2/d \boldsymbol{I}_d)]$, where, for each $c \in [K]$, $d\|\boldsymbol{\mu}_c\|_2^2 = \Theta^{1/d}(1)$, $p_c \in [0, 1]$ and the expectation is over $\sigma_c \sim \varrho_c$, generic distribution density supported on $\mathbb{R}_+^*$. Moreover, $\sum_{c=1}^{K} p_c = 1$. For each $\boldsymbol{x}_i$, $i \in [n]$, the corresponding response $y_i \in \mathcal{Y}$ is drawn from a conditional law $P_0(y|\boldsymbol{\beta}_\star^\intercal \boldsymbol{x}_i)$ on $\mathcal{Y}$, with target weights $\boldsymbol{\beta}_\star \in \mathbb{R}^d$ having finite normalised norm $\beta_\star^2 := \lim_{d \to \infty} 1/d\|\boldsymbol{\beta}_\star\|_2^2$.*

**Assumption 4.2** (Predictor). *We consider the hypothesis class of generalised linear predictors $\mathcal{H} = \{f_{\boldsymbol{\beta}}(\boldsymbol{x}) := f(\boldsymbol{\beta}^\intercal \boldsymbol{x}), \boldsymbol{\beta} \in \mathbb{R}^d\}$, where $f : \mathbb{R} \to \mathcal{Y}$ is a generic activation function, and the weights $\boldsymbol{\beta} \in \mathbb{R}^d$ are obtained by minimising the following empirical risk:*

$$\hat{\boldsymbol{\beta}}_\lambda := \underset{\boldsymbol{\beta} \in \mathbb{R}^d}{\arg\min} \sum_{i=1}^{n} \ell(y_i, \boldsymbol{\beta}^\intercal \boldsymbol{x}_i) + \lambda r(\boldsymbol{\beta}), \qquad \lambda \in \mathbb{R}_+, \tag{18}$$

*for a convex loss function $\ell : \mathcal{Y} \times \mathbb{R} \to \mathbb{R}_+$ and a convex regularisation function $r : \mathbb{R} \to \mathbb{R}_+$.*

Under the further Assumption 2.3, in Appendix A it is shown that the following holds.

**Result 4.3** (High-dimensional asymptotics for the EMM). *Let us consider the independent random variables $\boldsymbol{\xi} \sim \mathcal{N}(\boldsymbol{0}, \boldsymbol{I}_d)$, and $z, \zeta \sim \mathcal{N}(0, 1)$. In the proportional asymptotic regime, the training and test error defined in Eq. 7 and associated with the M-estimator of the problem in Eq. 18 have a limit in probability as in Eq. 8, with*

$$\varepsilon_t(\alpha, \lambda, \beta_\star^2) = \sum_c p_c \int_{\mathcal{Y}} dy \, \mathbb{E}_{\sigma_c, \zeta} \left[ Z_0\left(y, \boldsymbol{\mu}_c^\intercal \boldsymbol{\beta}_\star + \tfrac{\sigma_c m}{\sqrt{q}}\zeta, \sigma_c^2 \beta_\star^2 - \tfrac{\sigma_c^2 m^2}{q}\right) \varphi\left(y, \sigma_c \sqrt{q}\zeta + v\sigma_c^2 f_c\right) \right],$$

$$\varepsilon_g(\alpha, \lambda, \beta_\star^2) = \sum_c p_c \int_{\mathcal{Y}} dy \int d\eta \int d\tau P_0(y|\tau) \mathbb{E}_{\sigma_c} \left[ \mathcal{N}\left( \binom{\tau}{\eta}; \binom{\boldsymbol{\mu}_c^\intercal \boldsymbol{\beta}_\star}{t_c}, \sigma_c^2 \binom{\beta_\star^2 \; m}{m \; q}\right) \right] \varphi(y, \eta). \tag{19}$$

*In particular, the estimation error is given by*

$$\varepsilon_{\text{est}} := \lim_{d \to +\infty} \tfrac{1}{d} \mathbb{E}_{\mathcal{D}}\left[ \|\hat{\boldsymbol{\beta}}_\lambda - \boldsymbol{\beta}_\star\|_2^2 \right] = \beta_\star^2 - 2m + q \tag{20}$$

*The required set of order parameters and proximals can be found by self-consistently solving the equations below*

$$v = \frac{1}{d\sqrt{\hat{q}}} \mathbb{E}_{\boldsymbol{\xi}}[\boldsymbol{g}^\intercal \boldsymbol{\xi}], \qquad \hat{q}_c = \alpha p_c \int_{\mathcal{Y}} dy \, \mathbb{E}_{\sigma_c, \zeta} \left[ \sigma_c^2 Z_0\left(y, \boldsymbol{\mu}_c^\intercal \boldsymbol{\beta}_\star + \tfrac{\sigma_c m}{\sqrt{q}}\zeta, \sigma_c^2 \beta_\star^2 - \tfrac{\sigma_c^2 m^2}{q}\right) f_c^2 \right],$$

$$q = \tfrac{1}{d} \mathbb{E}_{\boldsymbol{\xi}}[\|\boldsymbol{g}\|_2^2], \qquad \hat{v} = -\alpha \sum_c p_c \int_{\mathcal{Y}} dy \, \mathbb{E}_{\sigma_c, \zeta} \left[ \sigma_c^2 Z_0\left(y, \boldsymbol{\mu}_c^\intercal \boldsymbol{\beta}_\star + \tfrac{\sigma_c m}{\sqrt{q}}\zeta, \sigma_c^2 \beta_\star^2 - \tfrac{\sigma_c^2 m^2}{q}\right) \partial_\omega f_c \right],$$

$$m = \tfrac{1}{d} \mathbb{E}_{\boldsymbol{\xi}}[\boldsymbol{g}^\intercal \boldsymbol{\beta}_\star], \qquad \hat{m}_c = \alpha p_c \int_{\mathcal{Y}} dy \, \mathbb{E}_{\sigma_c, \zeta} \left[ \sigma_c^2 \partial_\mu Z_0\left(y, \boldsymbol{\mu}_c^\intercal \boldsymbol{\beta}_\star + \tfrac{\sigma_c m}{\sqrt{q}}\zeta, \sigma_c^2 \beta_\star^2 - \tfrac{\sigma_c^2 m^2}{q}\right) f_c \right],$$

$$t_c = \mathbb{E}_{\boldsymbol{\xi}}[\boldsymbol{g}^\intercal \boldsymbol{\mu}_c], \qquad \hat{t}_c = \alpha p_c \int_{\mathcal{Y}} dy \, \mathbb{E}_{\sigma_c, \zeta} \left[ Z_0\left(y, \boldsymbol{\mu}_c^\intercal \boldsymbol{\beta}_\star + \tfrac{\sigma_c m}{\sqrt{q}}\zeta, \sigma_c^2 \beta_\star^2 - \tfrac{\sigma_c^2 m^2}{q}\right) f_c \right], \tag{21a}$$

*where, as before, $Z_0(y, \mu, V) := \mathbb{E}_z[P_0(y|\mu + \sqrt{V}z)]$ and*

$$\boldsymbol{g} := \arg\min_{\boldsymbol{\beta}} \left( \frac{\hat{v}\|\boldsymbol{\beta}\|_2^2}{2} - \sum_c \boldsymbol{\beta}^\intercal (\hat{m}_c \boldsymbol{\beta}_\star + d\hat{t}_c \boldsymbol{\mu}_c) - \sum_c \sqrt{\hat{q}_c} \boldsymbol{\xi}^\intercal \boldsymbol{\beta} + \lambda r(\boldsymbol{\beta}) \right) \in \mathbb{R}^d,$$

$$f_c := \arg\min_u \left[ \frac{vu^2\sigma_c^2}{2} + \ell\left(y, t_c + \sigma_c \sqrt{q}\zeta + \sigma_c^2 vu\right) \right] \in \mathbb{R}. \tag{21b}$$

The previous set of equations covers a wide range of distributions and possibly any power-law tail decay. The generality of this setting relies on the fact that any distribution can be approximated by a possibly uncountable superposition of Gaussians (Nestoridis et al., 2011; Alspach and Sorenson, 1972; Ghosh and Ramamoorthi, 2006). Moreover, in Appendices B & B.1 the following universality result, generalising results of Gerace et al. (2022) and Pesce et al. (2023) in the Gaussian setting, is given.

**Result 4.4** (Universality for uncorrelated target). *Assume that, for all $c \in [K]$, $\lim_{d \to +\infty} \boldsymbol{\mu}_c^\intercal \boldsymbol{\beta}_\star = 0$ and that the labels are generated according to the model in Eq. 1a, the noise $\eta$ having even distribution and zero mean. Then, if the loss $\ell$ is even, the asymptotic training and generalisation errors are the same as if the covariates had distribution $p(\boldsymbol{x}) = \mathbb{E}[\mathcal{N}(\boldsymbol{x}; \boldsymbol{0}, \sigma/d \boldsymbol{I}_d)]$ with $\sigma \sim \sum_{c=1}^{K} p_c \varrho_c$. Furthermore, if $\ell(y, t) = \frac{1}{2}(y - t)^2$ and $r(\boldsymbol{x}) = \frac{1}{2}\|\boldsymbol{x}\|_2^2$, under the assumption that $\mathbb{E}[\eta^2] < +\infty$, then the training loss is*

$$\lim_{\lambda \to 0^+} \tfrac{1}{2n} \sum_{i=1}^{n} (y_i - \boldsymbol{\beta}_\lambda^\intercal \boldsymbol{x}_i)^2 \xrightarrow[\alpha = n/d = \Theta(1)]{d \to +\infty} \frac{\mathbb{E}[\eta^2]}{2} \left(1 - \frac{1}{\alpha}\right)_+, \qquad (x)_+ := x\theta(x), \tag{22}$$

*for any distribution of the random variables $\sigma_c$.*

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

# Supplementary Information

## A    Replica derivation of the fixed-point equations

In this Appendix, we will derive the fixed point equations for the order parameters following the analysis by Loureiro et al. (2021); Pesce et al. (2023); Adomaityte et al. (2023) in the most general setting discussed in Result 4.3: the case in Result 2.4 is obtained by fixing $K = 1$ and $\boldsymbol{\mu}_1 \equiv \mathbf{0}$ below, and by assuming a ridge regularisation. The dataset $\mathcal{D} \coloneqq \{(\boldsymbol{x}_i, y_i)\}_{i \in [n]}$ consists of $n$ independent datapoints $\boldsymbol{x}_i \in \mathbb{R}^d$ each associated to a label $y_i \in \mathcal{Y}$. The elements of the dataset are independently generated by using a law $P(\boldsymbol{x}, y)$ which we assume can be put in the form of a superstatistical mixture model (SMM) involving $K$ clusters $\mathcal{C} = \{1, \ldots, K\}$,

$$P(\boldsymbol{x}, y) \equiv P_0(y|\boldsymbol{\beta}_\star^\mathsf{T} \boldsymbol{x}) \sum_{c \in \mathcal{C}} p_c \mathbb{E}_{\sigma_c} \left[ \mathcal{N}\left(\boldsymbol{x}; \boldsymbol{\mu}_c, \sigma_c^2/d \boldsymbol{I}_d\right) \right], \tag{23}$$

and $P_0(\bullet|\tau)$ is the distribution of the scalar label $y$ produced via the "teacher" $\boldsymbol{\beta}_\star$. In the following, we assume that $\beta_\star^2 = 1/d \|\boldsymbol{\beta}_\star\|_2^2 = \Theta(1)$. In the equation above, $\forall c \in \mathcal{C}$, $p_c \in [0, 1]$ and $\boldsymbol{\mu}_c \in \mathbb{R}^d$ with $\|\boldsymbol{\mu}_c\|_2^2 = \Theta(1/d)$. It is assumed that $\sum_c p_c = 1$. The expectation is intended over $\sigma_c$, a positive random variable with density $\varrho_c$. We will perform our regression task searching for a set of *weights* $\hat{\boldsymbol{\beta}}_\lambda$, that will allow us to construct an estimator via a certain classifier $f \colon \mathbb{R} \to \mathcal{Y}$:

$$\boldsymbol{x} \mapsto f(\hat{\boldsymbol{\beta}}_\lambda^\mathsf{T} \boldsymbol{x}) = y, \tag{24}$$

which will provide us with our prediction for a datapoint $\boldsymbol{x}$. The weights will be chosen by minimising an empirical risk function in the form

$$\mathcal{R}(\boldsymbol{\beta}) \equiv \sum_{\nu=1}^n \ell\left(y_i, \boldsymbol{\beta}^\mathsf{T} \boldsymbol{x}_i\right) + \lambda r(\boldsymbol{\beta}), \tag{25}$$

i.e., they are given by

$$\hat{\boldsymbol{\beta}}_\lambda \coloneqq \arg\min_{\boldsymbol{\beta} \in \mathbb{R}^d} \mathcal{R}(\boldsymbol{\beta}). \tag{26}$$

We will assume that $\ell$ is a convex loss function with respect to its second argument, and $r$ is a strictly convex regularisation function: the parameter $\lambda \geq 0$ will tune the strength of the regularisation. Note that this setting is slightly more general than the one given in the main text. The starting point is to reformulate the problem as an optimisation problem by introducing a Gibbs measure over the parameters $\boldsymbol{\beta}$ depending on a positive parameter $\upbeta$,

$$\mu_\upbeta(\boldsymbol{\beta}) \propto \mathrm{e}^{-\upbeta \mathcal{R}(\boldsymbol{\beta})} = \underbrace{\mathrm{e}^{-\upbeta r(\boldsymbol{\beta})}}_{P_w} \prod_{i=1}^n \underbrace{\exp\left[-\upbeta \ell\left(y_i, \boldsymbol{\beta}^\mathsf{T} \boldsymbol{x}_i\right)\right]}_{P_\ell}, \tag{27}$$

so that, in the $\upbeta \to +\infty$ limit, the Gibbs measure concentrates on $\hat{\boldsymbol{\beta}}_\lambda$. The functions $P_y$ and $P_w$ can be interpreted as (unnormalised) likelihood and prior distribution respectively. Our analysis will go through the computation of the average free energy density associated with such Gibbs measure in a specific proportional limit, i.e.,

$$f_\upbeta \coloneqq - \lim_{\substack{n,d \to +\infty \\ n/d = \alpha}} \mathbb{E}_\mathcal{D}\left[\frac{\ln \mathcal{Z}_\upbeta}{d\upbeta}\right] = \lim_{\substack{n,d \to +\infty \\ n/d = \alpha}} \lim_{s \to 0} \frac{1 - \mathbb{E}_\mathcal{D}[\mathcal{Z}_\upbeta^s]}{sd\upbeta}, \tag{28}$$

where $\mathbb{E}_\mathcal{D}[\bullet]$ is the average over the training dataset, and we have introduced the partition function

$$\mathcal{Z}_\upbeta \coloneqq \int \mathrm{e}^{-\upbeta \mathcal{R}(\boldsymbol{\beta})} \, \mathrm{d}\boldsymbol{\beta}. \tag{29}$$

### A.1    Replica approach.

In our replica approach, we need to evaluate

$$\mathbb{E}_\mathcal{D}[\mathcal{Z}_\upbeta^s] = \prod_{a=1}^s \int \mathrm{d}\boldsymbol{\beta}^a P_w(\boldsymbol{\beta}^a) \left(\mathbb{E}_{(\boldsymbol{x}, y)}\left[\prod_{a=1}^s P_\ell(y|\boldsymbol{x}^\mathsf{T} \boldsymbol{\beta}^a)\right]\right)^n. \tag{30}$$

Let us take the inner average introducing a new set of *local fields* $\eta^a$ and $\tau$,

$$\mathbb{E}_{(\boldsymbol{x},y)}\left[\prod_{a=1}^{s} P_\ell(y\big|\boldsymbol{x}^\intercal\boldsymbol{\beta}^a)\right] = \sum_c p_c \mathbb{E}_{\sigma_c}\left[\int_{\mathcal{Y}} \mathrm{d}y \int_{\mathbb{R}^d} \mathrm{d}\boldsymbol{x}\, P_0(y|\boldsymbol{x}^\intercal\boldsymbol{\beta}_\star)\mathcal{N}(\boldsymbol{x};\boldsymbol{\mu}_c,\sigma_c^2/d\boldsymbol{I}_d)\prod_{a=1}^{s} P_\ell(y|\boldsymbol{x}^\intercal\boldsymbol{\beta}^a)\right]$$

$$= \sum_c p_c \mathbb{E}_{\sigma_c}\left[\int \mathrm{d}\boldsymbol{\eta}\int \mathrm{d}\tau \int_{\mathcal{Y}} \mathrm{d}y\, P_0(y|\tau)\prod_{a=1}^{s} P_\ell(y|\eta^a)\mathcal{N}\left(\binom{\tau}{\boldsymbol{\eta}};\binom{\boldsymbol{\mu}_c^\intercal\boldsymbol{\beta}_\star}{\boldsymbol{\mu}_c^\intercal\boldsymbol{\beta}^a},\frac{\sigma_c^2}{d}\begin{pmatrix} d\beta_\star^2 & \boldsymbol{\beta}_\star^\intercal\boldsymbol{\beta}^b \\ \boldsymbol{\beta}_\star^\intercal\boldsymbol{\beta}^a & \boldsymbol{\beta}^{a\intercal}\boldsymbol{\beta}^b \end{pmatrix}\right)\right]. \quad (31)$$

We can write then

$$\mathbb{E}_{\mathcal{D}}[\mathcal{Z}_\beta^s] = \prod_{a=1}^{s}\int \mathrm{d}\boldsymbol{\beta}^a P_w(\boldsymbol{\beta}^a)\times$$

$$\left(\sum_c p_c \mathbb{E}_{\sigma_c}\left[\int \mathrm{d}\boldsymbol{\eta}\int \mathrm{d}\tau \int_{\mathcal{Y}} \mathrm{d}y\, P_0(y|\tau)\prod_{a=1}^{s} P_\ell(y|\eta^a)\mathcal{N}\left(\binom{\tau}{\boldsymbol{\eta}};\binom{\boldsymbol{\mu}_c^\intercal\boldsymbol{\beta}_\star}{\boldsymbol{\mu}_c^\intercal\boldsymbol{\beta}^a},\frac{\sigma_c^2}{d}\begin{pmatrix} d\beta_\star^2 & \boldsymbol{\beta}_\star^\intercal\boldsymbol{\beta}^b \\ \boldsymbol{\beta}_\star^\intercal\boldsymbol{\beta}^a & \boldsymbol{\beta}^{a\intercal}\boldsymbol{\beta}^b \end{pmatrix}\right)\right]\right)^n$$

$$= \prod_c\left(\prod_{a\leq b}\iint \mathcal{D}\boldsymbol{Q}^{ab}\mathcal{D}\hat{\boldsymbol{Q}}^{ab}\right)\left(\prod_a \int \mathcal{D}\boldsymbol{M}^a\mathcal{D}\hat{\boldsymbol{M}}^a\right)\left(\prod_a \int \mathrm{d}t^a\,\mathrm{d}\hat{t^a}\right)\mathrm{e}^{-d\beta\Phi^{(s)}}. \quad (32)$$

In the equation above we introduced the *order parameters*

$$Q_c^{ab} = \frac{\sigma_c^2}{d}\boldsymbol{\beta}^{a\intercal}\boldsymbol{\beta}^b \in \mathbb{R}, \quad a,b=1,\ldots,s, \quad (33)$$

$$M_c^a = \frac{\sigma_c^2}{d}\boldsymbol{\beta}_\star^\intercal\boldsymbol{\beta}^a \in \mathbb{R}, \quad a=1,\ldots,s, \quad (34)$$

$$t_c^a = \boldsymbol{\mu}_c^\intercal\boldsymbol{\beta}^a \in \mathbb{R}, \quad a=1,\ldots,s, \quad (35)$$

whilst the integration is over all possible order parameters, $Q_c^{ab}$ and $m_c^a$ to be intended as functions of $\sigma_c$. In the equation, we have also denoted the replicated free-energy

$$\beta\Phi^{(s)}(\boldsymbol{Q},\boldsymbol{M},\hat{\boldsymbol{Q}},\hat{\boldsymbol{M}}) = \sum_c\sum_a \mathbb{E}_{\sigma_c}[\hat{M}_c^a M_c^a] + \sum_c\sum_{a\leq b}\mathbb{E}_{\sigma_c}[\hat{Q}_c^{ab}Q_c^{ab}] + \frac{1}{d}\sum_{c,a}\hat{t}_c^a t_c^a$$

$$-\frac{1}{d}\ln\prod_{a=1}^{s}\int P_w(\boldsymbol{\beta}^a)\mathrm{d}\boldsymbol{\beta}^a\prod_c\exp\left(\sum_{a\leq b}\mathbb{E}_{\sigma_c}[\sigma_c^2\hat{Q}_c^{ab}]\boldsymbol{\beta}^{a\intercal}\boldsymbol{\beta}^b+\sum_a\mathbb{E}_{\sigma_c}[\sigma_c^2\hat{M}_c^a]\boldsymbol{\beta}^{a\intercal}\boldsymbol{\beta}_\star+\sum_a\hat{t}_c^a\boldsymbol{\beta}^{a\intercal}\boldsymbol{\mu}_c\right)$$

$$-\alpha\ln\sum_c p_c\mathbb{E}_{\sigma_c}\left[\int \mathrm{d}\boldsymbol{\eta}\int \mathrm{d}\tau \int_{\mathcal{Y}} \mathrm{d}y\, P_0(y|\tau)\prod_{a=1}^{s} P_\ell(y|\eta^a)\,\mathcal{N}\left(\binom{\tau}{\boldsymbol{\eta}};\binom{t_c^0}{t_c^a}\begin{pmatrix} \sigma_c^2\beta_\star^2 & M_c^b \\ M_c^a & Q_c^{ab} \end{pmatrix}\right)\right], \quad (36)$$

where, for the sake of brevity, $t_c^0 := \boldsymbol{\mu}_c^\intercal\boldsymbol{\beta}_\star$. At this point, the free energy $f_\beta$ should be computed functionally extremisizing with respect to all the order parameters by virtue of the Laplace approximation,

$$f_\beta = \lim_{s\to 0}\operatorname*{Extr}_{\substack{\boldsymbol{M},\hat{\boldsymbol{M}},\boldsymbol{t}\\ \boldsymbol{Q},\hat{\boldsymbol{Q}},\hat{\boldsymbol{t}}}}\frac{\Phi^{(s)}(\boldsymbol{Q},\boldsymbol{M},\hat{\boldsymbol{Q}},\hat{\boldsymbol{M}},\boldsymbol{t},\hat{\boldsymbol{t}})}{s}. \quad (37)$$

**Replica symmetric ansatz.** Before taking the $s\to 0$ limit we make the replica symmetric assumptions

$$\begin{aligned} Q_c^{aa} &= \begin{cases} R_c, & a=b \\ Q_c & a\neq b \end{cases} & \hat{Q}_c^{aa} &= \begin{cases} -\frac{1}{2}\hat{R}_c, & a=b \\ \hat{Q}_c & a\neq b \end{cases} \\ M_c^a &= M_c & \hat{M}_c^a &= \hat{M}_c \quad \forall a \\ t_c^a &= t_c & \hat{t}_c^a &= \hat{t}_c \quad \forall a \end{aligned} \quad (38)$$

If we denote $V_c := R_c - Q_c$ we obtain, after some work we obtain

$$\ln\sum_c p_c \mathbb{E}_{\sigma_c}\left[\int d\boldsymbol{\eta}\int d\tau\int_{\mathcal{y}} dy\, P_0(y|\tau)\prod_{a=1}^s P_\ell(y|\eta^a)\,\mathcal{N}\left(\begin{pmatrix}\tau\\\boldsymbol{\eta}\end{pmatrix};\begin{pmatrix}t_c^0\\t_c\mathbf{1}_s\end{pmatrix},\begin{pmatrix}\sigma_c^2\beta_\star^2 & M_c\mathbf{1}_s^\intercal\\M_c\mathbf{1}_s & Q_c\mathbf{1}_{s\times s}\end{pmatrix}\right)\right]$$

$$=s\sum_c p_c\mathbb{E}_{\sigma_c,\zeta}\left[\int_{\mathcal{y}}dy\,Z_0\left(y,t_c^0+\frac{M_c\zeta}{\sqrt{Q_c}},\sigma_c^2\beta_\star^2-\frac{M_c^2}{Q_c}\right)\ln Z_\ell\left(y,t_c+\sqrt{Q_c}\zeta,V_c\right)\right]+o(s),\quad (39)$$

with $\zeta\sim\mathcal{N}(0,1)$ is normally distributed and we have introduced the function

$$Z_\bullet(y,\mu,V):=\int\frac{d\tau P_\bullet(y|\tau)}{\sqrt{2\pi V}}\,e^{-\frac{(\tau-\mu)^2}{2V}},\qquad \bullet\in\{0,\ell\}. \quad (40)$$

On the other hand, denoted by $\hat{V}_c=\hat{R}_c+\hat{Q}_c$, and introducing $\hat{q}_c:=\mathbb{E}_{\sigma_c}[\sigma_c^2\hat{Q}_c]$, $\hat{v}_c:=\mathbb{E}_{\sigma_c}[\sigma_c^2\hat{V}_c]$, and $\hat{m}_c:=\mathbb{E}_{\sigma_c}[\sigma_c^2\hat{M}_c]$

$$\frac{1}{d}\ln\prod_{a=1}^s\left(\int P_w(\boldsymbol{\beta}^a)d\boldsymbol{\beta}^a\prod_c e^{-\frac{\hat{v}_c}{2}\|\boldsymbol{\beta}^a\|_2^2+\boldsymbol{\beta}^{a\intercal}(\hat{m}_c\boldsymbol{\beta}_\star+\hat{t}_c\boldsymbol{\mu}_c)}\prod_{b,c}e^{\frac{1}{2}\hat{q}_c\boldsymbol{\beta}^{a\intercal}\boldsymbol{\beta}^b}\right)=$$

$$=\frac{s}{d}\mathbb{E}_{\boldsymbol{\xi}}\ln\left[\int P_w(\boldsymbol{\beta})d\boldsymbol{\beta}\prod_c\exp\left(-\frac{\hat{v}_c\|\boldsymbol{\beta}\|_2^2}{2}+\boldsymbol{\beta}^\intercal(\hat{m}_c\boldsymbol{\beta}_\star+\hat{t}_c\boldsymbol{\mu}_c)+\sqrt{\hat{q}_c}\boldsymbol{\xi}^\intercal\boldsymbol{\beta}\right)\right]+o(s).\quad (41)$$

In the expression above we have introduced $\boldsymbol{\xi}\sim\mathcal{N}(\mathbf{0},\boldsymbol{I}_d)$. Therefore, the (replicated) *replica symmetric* free-energy is given by

$$\lim_{s\to 0}\frac{\beta}{s}\Phi_{\mathrm{RS}}^{(s)}=\frac{1}{d}\sum_c\hat{t}_c t_c+\sum_c\hat{M}_c M_c+\frac{\sum_c\mathbb{E}_{\sigma_c}\left[\hat{V}_c Q_c-\hat{Q}_c V_c-\hat{V}_c V_c\right]}{2}-\alpha\beta\Psi_\ell(M,Q,V)-\beta\Psi_w(\hat{m},\hat{q},\hat{v})$$

$$(42)$$

where we have defined two contributions

$$\Psi_\ell(M,Q,V):=\frac{1}{\beta}\sum_c p_c\mathbb{E}_{\sigma_c,\zeta}\left[\int_{\mathcal{y}}dy\,Z_0\left(y,t_c^0+\frac{M_c\zeta}{\sqrt{Q_c}},\sigma_c^2\beta_\star^2-\frac{M_c^2}{Q_c}\right)\ln Z_\ell\left(y,t_c+\sqrt{Q_c}\zeta,V_c\right)\right],$$

$$\Psi_w(\hat{m},\hat{q},\hat{v}):=\frac{1}{\beta d}\mathbb{E}_{\boldsymbol{\xi}}\ln\left[\int P_w(\boldsymbol{\beta})d\boldsymbol{\beta}\prod_c\exp\left(-\frac{\hat{v}_c\|\boldsymbol{\beta}\|_2^2}{2}+\boldsymbol{\beta}^\intercal(\hat{m}_c\boldsymbol{\beta}_\star+\hat{t}_c\boldsymbol{\mu}_c)+\sqrt{\hat{q}_c}\boldsymbol{\xi}^\intercal\boldsymbol{\beta}\right)\right].$$

$$(43)$$

Note that we have separated the contribution coming from the chosen loss (the so-called *channel* part $\Psi_\ell$) from the contribution depending on the regularisation (the *prior* part $\Psi_w$). To write down the saddle-point equations in the $\beta\to+\infty$ limit, let us first rescale our order parameters as $\hat{M}_c\mapsto\beta\hat{M}_c$, $\hat{t}_c\mapsto d\beta\hat{t}_c$, $\hat{Q}_c\mapsto\beta^2\hat{Q}_c$, $\hat{V}_c\mapsto\beta\hat{V}_c$ and $V_c\mapsto\beta^{-1}V_c$. Also, for future convenience, let us rescale $Q_c\mapsto\sigma_c^2 q_c$, $M_c\mapsto\sigma_c^2 m_c$, $V_c\mapsto\sigma_c^2 v_c$. For $\beta\to+\infty$ the channel part is

$$\Psi_\ell(m,q,v,t)=$$

$$=-\sum_c p_c\mathbb{E}_{\sigma_c,\zeta}\left[\int_{\mathcal{y}}dy\,Z_0\left(y,t_c^0+\frac{\sigma_c m\zeta}{\sqrt{q_c}},\sigma_c^2\beta_\star^2-\frac{\sigma_c^2 m_c^2}{q_c}\right)\left(\frac{(h_c-t_c-\sigma_c\sqrt{q_c}\zeta)^2}{2\sigma_c^2 v_c}+\ell(y,h_c)\right)\right].\quad (44)$$

where we have written $\Psi_\ell$ of a Moreau envelope, i.e., in terms of a proximal

$$h_c:=\arg\min_u\left[\frac{(u-\omega_c)^2}{2\sigma_c^2 v_c}+\ell(y,u)\right]\qquad\text{with }\omega_c=t_c+\sigma_c\sqrt{q_c}\zeta.\quad (45)$$

A similar expression can be obtained for $\Psi_w$. Introducing the proximal

$$\boldsymbol{g}=\arg\min_{\boldsymbol{\beta}}\left(\frac{\|\boldsymbol{\beta}\|_2^2\sum_c\hat{v}_c}{2}-\boldsymbol{\beta}^\intercal\sum_c\left(\hat{m}_c\boldsymbol{\beta}_\star+d\hat{t}_c\boldsymbol{\mu}_c\right)-\boldsymbol{\xi}^\intercal\boldsymbol{\beta}\sum_c\sqrt{\hat{q}_c}+\lambda r(\boldsymbol{\beta})\right)\in\mathbb{R}^d\quad (46)$$

We can rewrite the prior contribution $\Psi_w$ as

$$\Psi_w(\hat{m},\hat{q},\hat{v},\hat{t})=-\frac{1}{d}\mathbb{E}_{\boldsymbol{\xi}}\left[\frac{\|\boldsymbol{g}\|_2^2}{2}\sum_c\hat{v}_c-\boldsymbol{g}^\intercal\sum_c\left(\hat{m}_c\boldsymbol{\beta}_\star+\hat{t}_c\boldsymbol{\mu}_c\right)-\boldsymbol{\xi}^\intercal\boldsymbol{g}\sum_c\sqrt{\hat{q}_c}+\lambda r(\boldsymbol{g})\right]\quad (47)$$

The parallelism between the two contributions is evident, aside from the different dimensionality of the involved objects. The replica symmetric free energy in the $\beta \to +\infty$ limit is computed by extremising with respect to the introduced order parameters,

$$f_{\mathrm{RS}} = \mathrm{Extr}\left[\sum_c \mathbb{E}_{\sigma_c}[\sigma_c^2 \hat{M}_c m_c] + \frac{1}{2}\sum_c \mathbb{E}_{\sigma_c}\left[\sigma_c^2\left(\hat{V}_c q_c - \hat{Q}_c v_c\right)\right] + \sum_c t_c \hat{t}_c \right.$$
$$\left. - \alpha \Psi_\ell(m, q, v, t) - \Psi_w(\hat{m}, \hat{q}, \hat{v}, \hat{t})\right]. \quad (48)$$

To do so, we have to write down a set of saddle-point equations and solve them.

**Saddle-point equations.**  The saddle-point equations are derived straightforwardly from the obtained free energy functionally extremising with respect to all parameters. It is easily seen that $v_c$, $q_c$ and $m_c$ are independent from $\sigma_c$, and that it is possible to reduce the number of variables by introducing $\hat{v} = \sum_c \hat{v}_c$, so that we can write

$$v_c = \frac{\mathbb{E}_{\boldsymbol{\xi}}[\boldsymbol{g}^{\mathsf{T}}\boldsymbol{\xi}]}{d\sqrt{\hat{q}_c}}, \quad (49a)$$

$$q = \frac{\mathbb{E}_{\boldsymbol{\xi}}[\|\boldsymbol{g}\|_2^2]}{d}, \quad (49b)$$

$$m = \frac{\mathbb{E}_{\boldsymbol{\xi}}[\boldsymbol{g}^{\mathsf{T}}\boldsymbol{\beta}_\star]}{d}, \quad (49c)$$

$$t_c = \mathbb{E}_{\boldsymbol{\xi}}[\boldsymbol{g}^{\mathsf{T}}\boldsymbol{\mu}_c]. \quad (49d)$$

and the remaining equations can be rewritten as

$$\hat{q}_c = \alpha p_c \int_{\mathcal{Y}} \mathrm{d}y\, \mathbb{E}_{\sigma_c, \zeta}\left[\sigma_c^2 Z_0\left(y, t_c^0 + \frac{\sigma_c m}{\sqrt{q}}\zeta, \sigma_c^2\beta_\star^2 - \frac{\sigma_c^2 m^2}{q}\right) f_c^2\right], \quad (50a)$$

$$\hat{v} = -\alpha \sum_c p_c \int_{\mathcal{Y}} \mathrm{d}y\, \mathbb{E}_{\sigma_c, \zeta}\left[\sigma_c^2 Z_0\left(y, t_c^0 + \frac{\sigma_c m}{\sqrt{q}}\zeta, \sigma_c^2\beta_\star^2 - \frac{\sigma_c^2 m^2}{q}\right)\partial_\omega f_c\right], \quad (50b)$$

$$\hat{m}_c = \alpha p_c \int_{\mathcal{Y}} \mathrm{d}y\, \mathbb{E}_{\sigma_c, \zeta}\left[\sigma_c^2 \partial_\mu Z_0\left(y, t_c^0 + \frac{\sigma_c m}{\sqrt{q}}\zeta, \sigma_c^2\beta_\star^2 - \frac{\sigma_c^2 m^2}{q}\right) f_c\right], \quad (50c)$$

$$\hat{t}_c = \alpha p_c \int_{\mathcal{Y}} \mathrm{d}y\, \mathbb{E}_{\sigma_c, \zeta}\left[Z_0\left(y, t_c^0 + \frac{\sigma_c m}{\sqrt{q}}\zeta, \sigma_c^2\beta_\star^2 - \frac{\sigma_c^2 m^2}{q}\right) f_c\right] \quad (50d)$$

with

$$f_c := \arg\min_u \left[\frac{\sigma_c^2 v_c u^2}{2} + \ell(y, \omega_c + \sigma_c^2 v_c u)\right], \qquad \omega_c = t_c + \sigma_c \sqrt{q}\zeta,$$

$$\boldsymbol{g} = \arg\min_{\boldsymbol{\beta}}\left(\frac{\|\boldsymbol{\beta}\|_2^2 \hat{v}}{2} - \boldsymbol{\beta}^{\mathsf{T}}\sum_c \left(\hat{m}_c\boldsymbol{\beta}_\star + d\hat{t}_c\boldsymbol{\mu}_c\right) - \boldsymbol{\xi}^{\mathsf{T}}\boldsymbol{\beta}\sum_c \sqrt{\hat{q}_c} + \lambda r(\boldsymbol{\beta})\right). \quad (51)$$

To obtain the replica symmetric free energy, therefore, the given set of equations has to be solved, and the result is then plugged in Eq. 48. The obtained saddle-point equations correspond to the ones given in the Result 4.3.

**Training and test errors.**  Let us show how to use the previous result to estimate the training loss and the generalisation error. Let us start from estimating

$$\varepsilon_\ell := \lim_{n \to +\infty} \frac{1}{n}\sum_{i=1}^n \ell(y_i, \hat{\boldsymbol{\beta}}_\lambda^{\mathsf{T}}\boldsymbol{x}_i). \quad (52)$$

The best way to proceed is to observe that

$$\varepsilon_\ell = -\lim_{\beta \to +\infty}\partial_\beta(\beta\Psi_\ell) = \sum_c p_c \int_{\mathcal{Y}} \mathrm{d}y\, \mathbb{E}_{\sigma_c, \zeta}\left[Z_0\left(y, t_c^0 + \frac{\sigma_c m}{\sqrt{q}}\zeta, \sigma_c^2\beta_\star^2 - \frac{\sigma_c^2 m^2}{q}\right)\ell(y, h_c)\right]. \quad (53)$$

This concentration result holds for a generic function $\varphi\colon \mathcal{Y} \times \mathbb{R} \to \mathbb{R}$, so that more generally, under Assumption 2.3,

$$\frac{1}{n}\sum_{i=1}^{n}\varphi(y_i,\hat{\boldsymbol{\beta}}_\lambda^\intercal \boldsymbol{x}_i)\xrightarrow[n,d\to+\infty]{\mathrm{P}}\sum_c p_c\int_{\mathcal{Y}}\mathrm{d}y\,\mathbb{E}_{\sigma_c,\zeta}\left[Z_0\left(y,t_c^0+\frac{\sigma_c m}{\sqrt{q}}\zeta,\sigma_c^2\beta_\star^2-\frac{\sigma_c^2 m^2}{q}\right)\varphi(y,h_c)\right]. \quad (54)$$

The expressions above hold in general, but, as anticipated, important simplifications can occur in the set of saddle-point equations Eq. 50 and Eq. 49 depending on the choice of the loss $\ell$ and of the regularization function $r$. The population's expectation (e.g., in the computation of the test error) of a performance function $\varphi\colon \mathcal{Y} \times \mathbb{R} \to \mathbb{R}$ is given instead by

$$\varepsilon_g \coloneqq \lim_{n\to+\infty}\mathbb{E}_{(y,\boldsymbol{x})}\left[\varphi(y,\hat{\boldsymbol{\beta}}_\lambda^\intercal \boldsymbol{x})\right], \quad (55)$$

where the expectation has to be taken on a newly sampled datapoint $(y,\boldsymbol{x}) \notin \mathcal{D}$. This expression can be rewritten as

$$\mathbb{E}_{y|\boldsymbol{x}}\left[\int \mathrm{d}\eta\, \varphi(y,\eta)\mathbb{E}_{\boldsymbol{x}}\left[\delta\left(\eta-\hat{\boldsymbol{\beta}}_\lambda^\intercal \boldsymbol{x}\right)\right]\right]$$
$$\xrightarrow[n,d\to+\infty]{\mathrm{P}}\int \mathrm{d}\eta \int \mathrm{d}\tau \int_{\mathcal{Y}}\mathrm{d}y\, P_0\left(y|\tau\right)\varphi(y,\eta)\sum_c p_c\mathbb{E}_{\sigma_c}\left[\mathcal{N}\left(\left(\begin{smallmatrix}\tau\\\eta\end{smallmatrix}\right);\left(\begin{smallmatrix}t_c^0\\t_c\end{smallmatrix}\right),\sigma_c^2\left(\begin{smallmatrix}\beta_\star^2 & m\\ m & q\end{smallmatrix}\right)\right)\right]. \quad (56)$$

This can be easily computed numerically once the order parameters are given. Finally, another relevant quantity for our investigations is the estimation mean-squared error

$$\varepsilon_{\mathrm{est}} \coloneqq \lim_{d\to+\infty}\frac{1}{d}\mathbb{E}_{\mathcal{D}}\left[\|\hat{\boldsymbol{\beta}}_\lambda - \boldsymbol{\beta}_\star\|_2^2\right] = \beta_\star^2 - 2m + q. \quad (57)$$

## A.2 Bayes-optimal estimator for $K = 1$

A derivation similar to the one discussed above can be repeated to obtain information on the statistical properties of the Bayes optimal estimator presented in Result 2.6. Given a dataset $\mathcal{D}$ of observation, we have that

$$P(\boldsymbol{\beta}|\mathcal{D}) = \frac{P(\boldsymbol{\beta})P(\mathcal{D}|\boldsymbol{\beta})}{\mathcal{Z}(\mathcal{D})} = \frac{P(\boldsymbol{\beta})}{\mathcal{Z}(\mathcal{D})}\prod_{i=1}^{n}P_0(y_i|\boldsymbol{\beta}^\intercal \boldsymbol{x}_i) \quad (58)$$

where $P(\boldsymbol{\beta})$ is the prior on the teacher that we assume to be Gaussian, $P(\boldsymbol{\beta}) = \mathcal{N}(\boldsymbol{\beta};\boldsymbol{0},\beta_\star^2\boldsymbol{I}_d)$, and

$$\mathcal{Z}(\mathcal{D})\coloneqq\int \mathrm{d}\boldsymbol{\beta}\, P(\boldsymbol{\beta})\prod_{i=1}^{n}P_0(y_i|\boldsymbol{\beta}^\intercal \boldsymbol{x}_i)=\frac{1}{(2\pi)^{n/2}}\int \mathrm{d}\boldsymbol{\beta}\exp\left(-\frac{\|\boldsymbol{\beta}\|_2^2}{2\beta_\star^2}+\sum_{i=1}^{n}\ln P_0(y_i|\boldsymbol{\beta}^\intercal \boldsymbol{x}_i)\right). \quad (59)$$

The calculation of $\mathcal{Z}(\mathcal{D})$ gives access in particular to the free entropy $\phi(\mathcal{D}) \coloneqq \lim_n \frac{1}{n}\ln \mathcal{Z}(\mathcal{D})$. The computation of $\phi$, which has an information-theoretical interpretation as mutual information, provides us the statistics of $\boldsymbol{\beta}$ according to the true posterior $P(\boldsymbol{\beta}|\mathcal{D})$. By assuming a concentration in the large $n$ limit, the calculation is performed on $\mathbb{E}_{\mathcal{D}}[\ln \mathcal{Z}(\mathcal{D})]$. Using the replica trick as before, the calculation can be repeated almost identically. For the sake of simplicity, we focus on the case in which only one cluster is present, centered in the origin. It is found that the statistics of a Bayes optimal estimator can be characterised therefore by an order parameter $\mathsf{q}$ satisfying a self-consistent equation not different from the one presented above (we will use below a different font to stress that we are currently analysing the Bayes optimal setting)

$$\hat{\mathsf{q}} = \alpha\int_{\mathcal{Y}}\mathrm{d}y\,\mathbb{E}_{\sigma,\zeta}\left[\sigma^2 Z_0(y,\mu,V)\left(\partial_\mu \ln Z_0(y,\mu,V)\right)^2\Big|_{\substack{\mu=\sigma\sqrt{\mathsf{q}}\zeta\\ V=\sigma^2(\beta_\star^2-\mathsf{q})}}\right], \quad \mathsf{q} = \frac{\beta_\star^4\hat{\mathsf{q}}}{1+\beta_\star^2\hat{\mathsf{q}}}. \quad (60)$$

with $Z_0(y,\mu,V) \coloneqq \mathbb{E}_z[P_0(y|\mu+\sqrt{V}z)]$ with $z \sim \mathcal{N}(0,1)$. We can then compute the Bayes optimal estimation error for the Bayes optimal estimator $\hat{\boldsymbol{\beta}}_{\mathrm{BO}} = \mathbb{E}_{\boldsymbol{\beta}|\mathcal{D}}[\boldsymbol{\beta}]$ as

$$\varepsilon_{\mathrm{BO}} = \lim_{d\to+\infty}\frac{1}{d}\|\boldsymbol{\beta}_\star - \hat{\boldsymbol{\beta}}_{\mathrm{BO}}\|_2^2 = \beta_\star^2 - \mathsf{q}. \quad (61)$$

# B ASYMPTOTIC RESULTS FOR RIDGE-REGULARISED LOSSES

Let us fix now $r(\boldsymbol{x}) = \frac{1}{2}\|\boldsymbol{x}\|_2^2$. In this case, the computation of $\Psi_w$ can be performed explicitly via a Gaussian integration, and the saddle-point equations can take a more compact form that is particularly suitable for a numerical solution. In particular, the prior proximal is found as

$$\boldsymbol{g} = \frac{\sum_c \left( \hat{m}_c \boldsymbol{\beta}_\star + d\hat{t}_c \boldsymbol{\mu}_c \right) + \sum_c \sqrt{\hat{q}_c} \boldsymbol{\xi}}{\lambda + \hat{v}} \tag{62}$$

so that the prior saddle-point equations obtained from $\Psi_w$ become

$$
\begin{aligned}
q &= \frac{1}{d}\left(\sum_c \frac{\hat{m}_c \boldsymbol{\beta}_\star + d\hat{t}_c \boldsymbol{\mu}_c}{\lambda + \hat{v}}\right)^2 + \left(\sum_c \frac{\sqrt{\hat{q}_c}}{\lambda + \hat{v}}\right)^2 \\
m &= \frac{\sum_c \left(\beta_\star^2 \hat{m}_c + t_c^0 \hat{t}_c\right)}{\lambda + \hat{v}} \\
v_c &= \frac{1}{\lambda + \hat{v}}\sum_{c'}\sqrt{\frac{\hat{q}_{c'}}{\hat{q}_c}} \\
t_c &= \frac{\sum_{c'}\left(\hat{t}_{c'}\mu_{c'c} + t_c^0 \hat{m}_{c'}\right)}{\lambda + \hat{v}},
\end{aligned}
\qquad
\begin{aligned}
\hat{q}_c &= \alpha p_c \int_{\mathcal{Y}} dy\, \mathbb{E}_{\sigma_c,\zeta}\left[\sigma_c^2 Z_0 f_c^2\right], \\
\hat{v} &= -\alpha \sum_c p_c \int_{\mathcal{Y}} dy\, \mathbb{E}_{\sigma_c,\zeta}\left[\sigma_c^2 Z_0 \partial_\omega f_c\right], \\
\hat{m}_c &= \alpha p_c \int_{\mathcal{Y}} dy\, \mathbb{E}_{\sigma_c,\zeta}\left[\sigma_c^2 \partial_\mu Z_0 f_c\right], \\
\hat{t}_c &= \alpha p_c \int_{\mathcal{Y}} dy\, \mathbb{E}_{\sigma_c,\zeta}[Z_0 f_c]
\end{aligned}
\tag{63}
$$

We have used the shorthand notation $Z_0 \equiv Z_0\left(y, t_c^0 + \frac{\sigma_c m}{\sqrt{q}}\zeta, \sigma_c^2\beta_\star^2 - \frac{\sigma_c^2 m^2}{q}\right)$ and $\mu_{cc'} := d\boldsymbol{\mu}_{c'}^\mathsf{T}\boldsymbol{\mu}_c$.

**Regression on one cloud: consistency** In the special case in which $|\mathcal{C}| = 1$ and the cloud is centered in the origin, we have $t_1 = \hat{t}_1 = 0$ and, dropping the subscript referring to the cluster,

$$
\begin{aligned}
q &= \frac{\beta_\star^2 \hat{m}^2 + \hat{q}}{(\lambda + \hat{v})^2} & \hat{q} &= \alpha \int_{\mathcal{Y}} dy\, \mathbb{E}_{\sigma,\zeta}\left[\sigma^2 Z_0\left(y, \frac{\sigma m}{\sqrt{q}}\zeta, \sigma^2\beta_\star^2 - \frac{\sigma^2 m^2}{q}\right) f^2\right], \\
m &= \frac{\beta_\star^2 \hat{m}}{\lambda + \hat{v}} & \hat{v} &= -\alpha \int_{\mathcal{Y}} dy\, \mathbb{E}_{\sigma,\zeta}\left[\sigma^2 Z_0\left(y, \frac{\sigma m}{\sqrt{q}}\zeta, \sigma^2\beta_\star^2 - \frac{\sigma^2 m^2}{q}\right) \partial_\omega f\right], \\
v &= \frac{1}{\lambda + \hat{v}}, & \hat{m} &= \alpha \int_{\mathcal{Y}} dy\, \mathbb{E}_{\sigma,\zeta}\left[\sigma^2 \partial_\mu Z_0\left(y, \frac{\sigma m}{\sqrt{q}}\zeta, \sigma^2\beta_\star^2 - \frac{\sigma^2 m^2}{q}\right) f\right],
\end{aligned}
\tag{64}
$$

where as usual $f := \arg\min_u \left[\frac{\sigma^2 v u^2}{2} + \ell(y, \omega + \sigma^2 v u)\right]$ and $\omega = \sigma\sqrt{q}\zeta$. Let us now perform the rescaling $v \mapsto \alpha v$, $\hat{q} \mapsto \alpha \hat{q}$, $\hat{m} \mapsto \alpha \hat{m}$, and $\hat{v} \mapsto \alpha \hat{v}$, where $v = O(1)$, $\hat{v} = O(1)$, $\hat{m} = O(1)$, $\hat{q} = O(1)$. Then, under these assumptions, in the $\alpha \to +\infty$ limit

$$
\begin{aligned}
q &= \frac{\beta_\star^2 \hat{m}^2}{\hat{v}^2} & \hat{q} &= \int_{\mathcal{Y}} dy\, \mathbb{E}_{\sigma,\zeta}\left[\sigma^2 Z_0\left(y, \frac{\sigma m}{\sqrt{q}}\zeta, \sigma^2\beta_\star^2 - \frac{\sigma^2 m^2}{q}\right) f^2\right], \\
m &= \frac{\beta_\star^2 \hat{m}}{\hat{v}} & \hat{v} &= -\int_{\mathcal{Y}} dy\, \mathbb{E}_{\sigma,\zeta}\left[\sigma^2 Z_0\left(y, \frac{\sigma m}{\sqrt{q}}\zeta, \sigma^2\beta_\star^2 - \frac{\sigma^2 m^2}{q}\right) \partial_\omega f\right], \\
v &= \frac{1}{\hat{v}}, & \hat{m} &= \int_{\mathcal{Y}} dy\, \mathbb{E}_{\sigma,\zeta}\left[\sigma^2 \partial_\mu Z_0\left(y, \frac{\sigma m}{\sqrt{q}}\zeta, \sigma^2\beta_\star^2 - \frac{\sigma^2 m^2}{q}\right) f\right],
\end{aligned}
\tag{65}
$$

Moreover, in the large $\alpha$ limit, $f = -\partial_\eta \ell(y, \eta)|_{\eta = \sigma\sqrt{q}\zeta}$. It follows that, independently from the adopted loss, the angle $\pi^{-1} \arccos \frac{m}{\beta_\star \sqrt{q}}$ between the estimator $\hat{\boldsymbol{\beta}}_\lambda$ and the teacher $\boldsymbol{\beta}_\star$ goes to zero as $\alpha \to +\infty$. In this limit, it is easily found that $\varepsilon_{\text{est}} \to \beta_\star^{-2}(m - \beta_\star^2)^2$, hence the estimator is consistent if $m \to \beta_\star^2$.

**Uncorrelated teachers: universality** To study the universality properties in the ridge setting, let us introduce two possible new assumptions.

**Assumption B.1.** *For all $c \in [K]$, $\lim_{d \to +\infty} t_c^0 = 0$.*

This assumption holds, for example, by assuming the centroids $\boldsymbol{\mu}_c \sim \mathcal{N}(\boldsymbol{0}, 1/d\boldsymbol{I}_d)$. It expresses in general the fact that the teacher $\boldsymbol{\beta}_\star$ is completely uncorrelated from the different centroids $\boldsymbol{\mu}_c$.

**Assumption B.2.** *The following symmetry properties hold*

$$P_0(y|\tau) = P_0(-y|-\tau), \qquad \ell(y,\eta) = \ell(-y,-\eta). \tag{66}$$

Under Assumption B.1 and Assumption B.2, $t_c = \hat{t}_c = 0 \ \forall c$ is a saddle-point solution of the equations 63. Indeed, if $\hat{t}_c = 0$ the prior equation implies $t_c = 0$. On the other hand, if $t_c = 0$, $\hat{t}_c = 0$ for parity reason (Pesce et al., 2023). We recover therefore in our setting the *mean universality* discussed by Pesce et al. (2023) in the Gaussian setting: the learning task is mean-independent and equivalent to one on $c$ clouds all centered in the origin, i.e., a problem obtained assuming $\boldsymbol{x} \sim \sum_c p_c \mathbb{E}_{\sigma_c}[\mathcal{N}(\boldsymbol{0}, \sigma_c/d\boldsymbol{I}_d)]$. Note that in the Gaussian setting (i.e., assuming $\rho_c(\sigma) = \delta(\sigma - \bar{\sigma}_c)$ for some fixed $\bar{\sigma}_c$ for each $c \in [C]$) Pesce et al. (2023) also observed that in the limit $\lambda \to 0^+$, *covariance universality* holds in the Gaussian case: $\varepsilon_g$ and $\varepsilon_\ell$ are independent on the covariance of the clouds. This fact does not extend to the case in which the distribution of $\sigma_c$ is not atomic (not even in the case in which $\sigma_c$ are all identically distributed), as it has been verified by Adomaityte et al. (2023).

## B.1 SQUARE LOSS

If we consider a square loss $\ell(y, \eta) = \frac{1}{2}(y - \eta)^2$, then an explicit formula for the proximal can be found, namely

$$f_c = \frac{y - \omega_c}{1 + v_c \sigma_c^2}, \qquad \omega_c = t_c + \sigma_c \sqrt{q}\zeta, \tag{67}$$

so that the second set of saddle-point equations Eq. 50 can be made even more explicit. Let us assume, that labels are generated according to the linear model in Eq. 1a, where the noise term $\eta_i$ has $\mathbb{E}[\eta_i] = 0$ and $\hat{\sigma}_0^2 := \mathbb{E}[\eta_i^2] < +\infty$. In this setting, the channel equations can be written as

$$\hat{q}_c = \alpha p_c \hat{\sigma}_0^2 \mathbb{E}_{\sigma_c}\left[\frac{\sigma_c^2}{(1 + v_c \sigma_c^2)^2}\right] + \alpha p_c \mathbb{E}_{\sigma_c}\left[\left(\frac{\sigma_c^2}{1 + v_c \sigma_c^2}\right)^2\right](\beta_\star^2 - 2m + q + (t_c^0 - t_c)^2), \tag{68a}$$

$$\hat{v} = \alpha \sum_c p_c \mathbb{E}_{\sigma_c}\left[\frac{\sigma_c^2}{1 + v_c \sigma_c^2}\right], \tag{68b}$$

$$\hat{m}_c = \alpha p_c \mathbb{E}_{\sigma_c}\left[\frac{\sigma_c^2}{1 + v_c \sigma_c^2}\right] \tag{68c}$$

$$\hat{t}_c = \alpha p_c (t_c^0 - t_c) \mathbb{E}_{\sigma_c}\left[\frac{1}{1 + v_c \sigma_c^2}\right]. \tag{68d}$$

In this setting the generalisation error becomes

$$\varepsilon_g := \lim_{d \to +\infty} \mathbb{E}\left[\left(y - \hat{\boldsymbol{\beta}}_\lambda^{\mathsf{T}} \boldsymbol{x}\right)^2\right] = \hat{\sigma}_0^2 + \sum_c p_c (t_c^0 - t_c)^2 + (\beta_\star^2 - 2m + q) \sum_c p_c \mathbb{E}_{\sigma_c}[\sigma_c^2], \tag{69}$$

which is finite if and only if $\mathbb{E}_{\sigma_c}[\sigma_c^2] < +\infty$ for all $c$. Note that the dependence on $\hat{\varrho}$ is through its second moment only. Observe that the possible power-law behavior of the noise on the label *does not* influence the generalisation performances, that only depends on the noise variance $\hat{\sigma}_0^2$. The training loss, on the other hand, is

$$\varepsilon_\ell := \lim_{d \to +\infty} \frac{1}{2n} \sum_{i=1}^n \left(y_i - \hat{\boldsymbol{\beta}}_\lambda^{\mathsf{T}} \boldsymbol{x}_i\right)^2 \xrightarrow{d \to +\infty} \frac{\hat{\sigma}_0^2}{2} \sum_c p_c \mathbb{E}_{\sigma_c}\left[\frac{1}{(1 + v_c \sigma_c^2)^2}\right] +$$

$$+ \sum_c \frac{p_c}{2}\left[\mathbb{E}_{\sigma_c}\left[\frac{1}{(1 + v_c \sigma_c^2)^2}\right](t_c^0 - t_c)^2 + \mathbb{E}_{\sigma_c}\left[\frac{\sigma_c^2}{(1 + v_c \sigma_c^2)^2}\right](\beta_\star^2 - 2m + q)\right]. \tag{70}$$

**Strong universality of $\varepsilon_\ell$ for $\lambda \to 0^+$**  We will show now that, under the Assumption B.1 (Assumption B.2 is satisfied in the setting under consideration), the strong universality of the training loss observed by Pesce et al. (2023) is preserved. Let us put ourselves in the

case of a single cluster centered in the origin (an assumption that is not restrictive, as shown above). In this case, let us introduce

$$S_v := \mathbb{E}_\sigma \left[ \frac{1}{1 + v\sigma^2} \right] \tag{71}$$

which can be interpreted in terms of the Stieljes transform of the random variable $\sigma^2$. The saddle-point equations are

$$
\begin{aligned}
\hat{q} &= -\alpha\hat{\sigma}_0^2 \partial_v S_v + \alpha(1 - S_v + v\partial_v S_v)\frac{\beta_\star^2 - 2m + q}{v^2}, &\quad q &= \frac{\hat{m}^2\beta_\star^2 + \hat{q}}{(\lambda + \hat{v})^2} \\
\hat{v} &= \alpha\frac{1 - S_v}{v}, &\quad m &= \frac{\beta_\star^2\hat{m}}{\lambda + \hat{v}} \\
\hat{m} &= \alpha\frac{1 - S_v}{v}, &\quad v &= \frac{1}{\lambda + \hat{v}}.
\end{aligned}
\tag{72}
$$

The training loss can be written as

$$\varepsilon_\ell = -\frac{\hat{\sigma}_0^2 \partial_v S_v}{2} - \frac{(\beta_\star^2 - 2m + q)\partial_v S_v}{2}. \tag{73}$$

In the limit $\lambda \to 0$,

$$x := \frac{\beta_\star^2 - 2m + q}{v} = \frac{(1 - S_v + v\partial_v S_v)x - v\partial_v S_v \hat{\sigma}_0^2}{1 - S_v} \Rightarrow x = \hat{\sigma}_0^2 \tag{74}$$

so that $\varepsilon_\ell = \frac{1}{2}S_v\hat{\sigma}_0^2$. The quantity $S_v$ can be extracted from the equation for $v$, as it has to satisfy, in the zero regularisation limit, $\alpha(1 - S_v) = 1 \Rightarrow S_v = 1 - \frac{1}{\alpha}$ which is a valid solution for $\alpha > 1$ only. As a result, we obtain a *universal* formula for the training loss

$$\varepsilon_\ell = \frac{\hat{\sigma}_0^2}{2}\left(1 - \frac{1}{\alpha}\right)_+, \qquad \text{where} \quad (x)_+ := x\theta(x). \tag{75}$$

Note that the formula above is valid for *any* distribution of $\sigma$, including distributions with no second moment.

**Generalisation error rate** — We conclude this section by extracting the generalisation error rate for $n \to +\infty$ and large but fixed $d$, i.e., for $\alpha \to +\infty$. For simplicity, let us focus, once again, on the case $K = 1$ and $\boldsymbol{\mu}_1 = \mathbf{0}$, corresponding to the fixed-point equations given in Eq. 72. Let us assume that $\hat{\sigma}_0^2 < +\infty$ and that $\sigma_0^2 := \mathbb{E}[\sigma^2] < +\infty$. From Eq. 72 $v$ satisfies the equation $\alpha(1 - S_v) = 1 - \lambda v$: as $S_v \in [0, 1]$ and $v > 0$, then for $\alpha \to +\infty$ we must have $S_v \to 1$ and $v \to 0$, so that for $\alpha \to +\infty$, $S_v = 1 - \frac{1}{\alpha} + O(\alpha^{-1})$. In this limit, therefore, by direct inspection of the fixed-point equations, $q \to \beta_\star^2$ and $m \to \beta_\star^2$ so that $\varepsilon_{\text{est}} \to 0$ and the estimator $\hat{\boldsymbol{\beta}}_\lambda$ is unbiased.

In the hypothesis that $\sigma_0^2$ is finite (i.e., $\varrho(\sigma) \sim \sigma^{-2a-1}$ with $a > 1$ for $\sigma \gg 1$), then, for small $v$, as $S_v \simeq 1 - v\sigma_0^2 + o(v)$, it is found that

$$q = \beta_\star^2 + \frac{\hat{\sigma}_0^2 - 2\beta_\star^2\lambda}{\sigma_0^2}\frac{1}{\alpha} + o\left(\frac{1}{\alpha}\right), \qquad m = \beta_\star^2 - \frac{\lambda\beta_\star^2}{\sigma_0^2}\frac{1}{\alpha} + o\left(\frac{1}{\alpha}\right), \qquad v = \frac{1}{\sigma_0^2\alpha} + o\left(\frac{1}{\alpha}\right), \tag{76}$$

which, together with our general formulas for $\varepsilon_{\text{est}}$, imply $\varepsilon_{\text{est}} \sim \alpha^{-1}$ for large $\alpha$.

On the other hand, let us consider the case in which $\varrho(\sigma) \sim \sigma^{-2a-1}$ for $\sigma \gg 1$, with $0 < a < 1$. In this case, $\sigma_0^2 = +\infty$ and $S_v$ has an expansion in the form $S_v = 1 - \tilde{\sigma}_0^2 v^a + O(v)$ for some finite positive quantity $\tilde{\sigma}_0^2$. Such asymptotic implies that $v \simeq (\tilde{\sigma}_0^2\alpha)^{-1/a}$ for $\alpha \gg 1$. By replacing this in the fixed point equations, it is found that $m = \beta_\star^2 - \lambda\beta_\star^2(\tilde{\sigma}_0^2\alpha)^{-1/a} + o(1/\alpha)$ and $q = \beta_\star^2 + [\hat{\sigma}_0^2 - 2\lambda\beta_\star^2](\tilde{\sigma}_0^2\alpha)^{-1/a} + o(1/\alpha)$, so that in the end $\varepsilon_{\text{est}} \sim \alpha^{-1/a}$.

The $a = 1$ case is marginal, as $S_v \simeq 1 + \tilde{\sigma}_0^2 v \log v$ for $\alpha \gg 1$ for some positive constant $\tilde{\sigma}_0^2$. Therefore $v = (\tilde{\sigma}_0^2\alpha \ln \alpha)^{-1}$. By consequence, for $\alpha \gg 1$ $m \simeq \beta_\star^2 - \lambda\beta_\star^2(\tilde{\sigma}_0^2\alpha \ln \alpha)^{-1}$ and $q \simeq \beta_\star^2 + [\hat{\sigma}_0^2 - 2\lambda\beta_\star^2](\tilde{\sigma}_0^2\alpha \ln \alpha)^{-1}$, so that $\varepsilon_{\text{est}} \sim (\alpha \ln \alpha)^{-1}$.

## B.2 Huber loss and robust regression in the presence of fat tails

### B.2.1 A model for the study of robustness

A toy model for the study of robustness has been introduced recently by Vilucchio et al. (2023). Here we will consider a more general setting to include the possibility of having fat tails. We consider the case of one cloud only, $K = 1$, so that $P(\boldsymbol{x}) = \mathbb{E}_\sigma[\mathcal{N}(\boldsymbol{x}; \mathbf{0}, \sigma^2/d\boldsymbol{I}_d)]$, and $P_0(y|\tau) = \mathbb{E}_{\hat{\sigma}}[\mathcal{N}(y; \hat{\eta}\tau, \hat{\sigma}^2)]$, where the expectation is taken over the joint distribution $\hat{\varrho}$ for the pair $(\hat{\eta}, \hat{\sigma})$ of (possibly correlated) random variables. Vilucchio et al. (2023) adopted, in particular, the distribution $\hat{\varrho}(\hat{\eta}, \hat{\sigma}) = \epsilon \delta_{\hat{\eta}, \hat{\eta}_{\mathrm{out}}} \delta_{\hat{\sigma}, \hat{\sigma}_{\mathrm{out}}} + (1 - \epsilon) \delta_{\eta, 1} \delta_{\hat{\sigma}, \hat{\sigma}_{\mathrm{in}}}$ for $\epsilon \in [0, 1]$, with $(\hat{\eta}_{\mathrm{out}}, \hat{\sigma}_{\mathrm{out}})$ referring to "outlier labels", and $(1, \hat{\sigma}_{\mathrm{in}})$ referring to "inlier labels". The general fixed-point equations can be adapted to this case quite easily. We assume, once again, a ridge regularisation. Here we comment on the fact that in this case, it can be interesting to consider, beyond the ERM estimator $\hat{\boldsymbol{\beta}}_\lambda$ and the Bayes-optimal estimator $\hat{\boldsymbol{\beta}}_{\mathrm{BO}}$, the estimator that minimises the (posterior-averaged) mean-squared test error

$$\hat{\boldsymbol{\beta}}_{g,\mathrm{BO}} = \arg\min_{\boldsymbol{\beta}} \mathbb{E}_{\hat{\boldsymbol{\beta}}|\mathcal{D}} \left[ \mathbb{E}_{(y,\boldsymbol{x})|\hat{\boldsymbol{\beta}}} \left[ (y - \boldsymbol{\beta}^\intercal \boldsymbol{x})^2 \right] \right] = \mathbb{E}[\hat{\eta}] \hat{\boldsymbol{\beta}}_{\mathrm{BO}}. \tag{77}$$

In the expression above, $\mathbb{E}_{(y,\boldsymbol{x})|\hat{\boldsymbol{\beta}}}$ expresses the fact that the pair $(y, \boldsymbol{x})$ has been generated with a teacher vector $\hat{\boldsymbol{\beta}}$, sampled by the posterior. Using the results on the Bayes optimal estimator, it is simple to derive the errors obtained by using $\hat{\boldsymbol{\beta}}_{g,\mathrm{BO}}$. Under the assumptions that $\sigma_0^2 := \mathbb{E}[\sigma^2] < +\infty$ and $\hat{\sigma}_0^2 := \mathbb{E}[\hat{\sigma}^2]$,

$$\varepsilon_{g,\mathrm{BO}} := \mathbb{E}_{(y,\boldsymbol{x})} \left[ (y - \boldsymbol{x}^\intercal \boldsymbol{\beta}_{g,\mathrm{BO}})^2 \right] = \sigma_0^2 \left( \beta_\star^2 \mathbb{E}[\hat{\eta}^2] - \mathbb{E}[\hat{\eta}]^2 \mathsf{q} \right) + \hat{\sigma}_0^2,$$

where $\mathsf{q}$ is provided by Eq. 60. As in the pure Gaussian case, by imposing the ansatz $\mathsf{q} = \beta_\star^2 - \frac{1}{\alpha} q_0 + \Theta(\alpha^{-2})$, and consequently $\hat{\mathsf{q}} = \alpha \hat{q}_0 + \Theta(1)$ for large $\alpha$, we can obtain

$$\frac{1}{q_0} = \hat{q}_0 = \int_{\mathcal{Y}} \mathrm{d}y \, \mathbb{E}_{\sigma,\zeta} \left[ \sigma^2 P_0(y|\omega) \left( \partial_\omega \ln P_0(y|\omega) \right)^2 \Big|_{\omega = \sigma\beta_\star\zeta} \right]. \tag{78}$$

In the $\alpha \to +\infty$ limit, then, $\mathsf{q} \to \rho$ and $\varepsilon_{\mathrm{est},\mathrm{BO}} := \lim_{d \to +\infty} \frac{1}{d} \mathbb{E}_{\mathcal{D}}[\|\hat{\boldsymbol{\beta}}_{g,\mathrm{BO}} - \boldsymbol{\beta}\|_2^2] = \frac{q_0}{\alpha} + \Theta(\alpha^{-2}) \to 0$. On the other hand, $\varepsilon_{g,\mathrm{BO}} = \hat{\sigma}_0^2 + \sigma_0^2 \beta_\star^2 \mathrm{Var}[\hat{\eta}] - \frac{\sigma_0^2 q_0}{\alpha} + \Theta(\alpha^{-2})$.

### B.2.2 Huber loss and its application

The Huber loss is a strongly convex loss depending on a tunable parameter $\delta \geq 0$ and is defined as

$$\ell_\delta(y, \eta) = \begin{cases} \frac{(y-\eta)^2}{2} & \text{if } |y - \eta| < \delta \\ \delta|y - \eta| - \frac{\delta^2}{2} & \text{otherwise.} \end{cases} \tag{79}$$

This loss is widely adopted in robust regression as it is less sensitive to outliers than the most commonly adopted square loss, and is associated with the following expression for the proximal

$$h_c = \omega_c + \frac{(y - \omega_c) v_c \sigma_c^2}{\max(\delta^{-1}|y - \omega_c|, 1 + v_c \sigma_c^2)} \Leftrightarrow f_c = \frac{y - \omega_c}{\max(\delta^{-1}|y - \omega_c|, 1 + v_c \sigma_c^2)}, \quad \omega_c = t_c + \sigma_c \sqrt{q} \zeta. \tag{80}$$

The prior equations are therefore the usual in Eq. 64. The channel equations are instead

$$\hat{m} = \alpha \mathbb{E} \left[ \frac{\sigma^2 \hat{\eta} \operatorname{erf} \chi}{1 + v\sigma^2} \right] \tag{81a}$$

$$\hat{q} = \alpha \mathbb{E} \left[ \frac{\sigma^2 \psi \operatorname{erf} \chi}{(1 + v\sigma^2)^2} + \sigma^2 \delta^2 (1 - \operatorname{erf} \chi) - \sqrt{\frac{2\psi}{\pi}} \frac{\sigma^2 \delta \, \mathrm{e}^{-\chi^2}}{1 + v\sigma^2} \right] \tag{81b}$$

$$\hat{v} = \alpha \mathbb{E} \left[ \frac{\sigma^2 \operatorname{erf} \chi}{1 + v\sigma^2} \right]. \tag{81c}$$

where the expectation is over all random variables involved in the expressions (namely, $\sigma$, $\hat{\sigma}$, and $\hat{\eta}$) and we used the short-hand notation

$$\psi := \hat{\sigma}^2 + \sigma^2(\hat{\eta}^2\beta_\star^2 - 2\hat{\eta}m + q), \qquad \chi := \frac{\delta(1 + v\sigma^2)}{\sqrt{2\psi}} \tag{82}$$

Note that we recover the expressions obtained for the square loss for $\delta \to +\infty$.

With the usual notation convention $\hat{\sigma}_0^2 := \mathbb{E}[\hat{\sigma}^2]$ and $\sigma_0^2 := \mathbb{E}[\sigma^2]$, the estimation error is given by the general formula in Eq. 57, whereas the generalisation error is

$$\varepsilon_g := \mathbb{E}\left[(y - \hat{\boldsymbol{\beta}}_\lambda^\mathsf{T}\boldsymbol{x})^2\right] = \hat{\sigma}_0^2 + (\beta_\star^2\mathbb{E}[\hat{\eta}^2] - 2\mathbb{E}[\hat{\eta}]m + q)\sigma_0^2, \tag{83}$$

$\varepsilon_g$ being finite if $\sigma^2 < +\infty$, $\hat{\sigma}_0^2 < +\infty$ and $\mathbb{E}[\hat{\eta}] < +\infty$. We aim now at extrapolating the large-$\alpha$ behavior of such errors and at studying the consistency of $\hat{\boldsymbol{\beta}}_\lambda$ with respect to the Bayes optimal estimators discussed in Section A.2. To do so, we rescale $\hat{m} \mapsto \alpha\hat{m}$, $\hat{v} \mapsto \alpha\hat{v}$, $v \mapsto \alpha^{-1}v$ and $\hat{q} \mapsto \alpha\hat{q}$. We also assume that $\lambda \mapsto \lambda + \alpha\lambda'$ (the role of $\lambda' \neq 0$ will be clear in the following). The set of fixed point equations become, for $\alpha \to +\infty$

$$\hat{m} = \mathbb{E}\left[\sigma^2\hat{\eta}\operatorname{erf}\bar{\chi}\right] \qquad\qquad q = \frac{\beta_\star^2\hat{m}^2}{(\lambda' + \hat{v})^2}$$

$$\hat{q} = \mathbb{E}\left[\sigma^2\psi\operatorname{erf}\bar{\chi} + \sigma^2\delta^2(1 - \operatorname{erf}\bar{\chi}) - \sqrt{\frac{2\psi}{\pi}}\sigma^2\delta\,\mathrm{e}^{-\bar{\chi}^2}\right] \qquad m = \frac{\beta_\star^2\hat{m}}{\lambda' + \hat{v}} \quad , \qquad \bar{\chi} := \frac{\delta}{\sqrt{2\psi}}.$$

$$\hat{v} = \mathbb{E}\left[\sigma^2\operatorname{erf}\bar{\chi}\right]. \qquad\qquad v = \frac{1}{\lambda' + \hat{v}}$$

$$\tag{84}$$

In this limit, as $\beta_\star^2 q = m^2$, $\psi = \hat{\sigma}_0^2 + \frac{\sigma_0^2}{\beta_\star^2}(m - \beta_\star^2\hat{\eta})^2$, so that

$$\varepsilon_{\text{est}} = \frac{(m - \beta_\star^2)^2}{\beta_\star^2}, \qquad \varepsilon_g = \hat{\sigma}_0^2 + \sigma^2\frac{\mathbb{E}[(m - \hat{\eta}\beta_\star^2)^2]}{\beta_\star^2}. \tag{85}$$

It is possible to choose $\lambda'$ so that $\lim_{\alpha \to +\infty}\varepsilon_g = \lim_{\alpha \to +\infty}\varepsilon_g^{\text{BO}}$, i.e.,

$$\hat{\sigma}_0^2 + \sigma_0^2\frac{\mathbb{E}[(m - \hat{\eta}\beta_\star^2)^2]}{\beta_\star^2} = \hat{\sigma}_0^2 + \sigma_0^2\beta_\star^2\operatorname{Var}[\hat{\eta}] \Rightarrow m = \beta_\star^2\mathbb{E}[\hat{\eta}]. \tag{86}$$

We can try to satisfy this condition by tuning properly $\lambda'$, under the constraint that $\lambda' \geq 0$. We can write in particular

$$\lambda' = \frac{\hat{m}}{\mathbb{E}[\hat{\eta}]} - \hat{v} = \frac{\mathbb{E}[\sigma^2\hat{\eta}\operatorname{erf}\bar{\chi}] - \mathbb{E}[\hat{\eta}]\mathbb{E}[\sigma^2\operatorname{erf}\bar{\chi}]}{\mathbb{E}[\hat{\eta}]} \geq 0 \Rightarrow \mathbb{E}[\sigma^2(\hat{\eta} - \mathbb{E}[\hat{\eta}])\operatorname{erf}\bar{\chi}] \geq 0 \tag{87}$$

to be computed with

$$\bar{\chi} \equiv \frac{\delta}{\sqrt{2\psi_g}}, \quad \psi_g = \hat{\sigma}_0^2 + \sigma_0^2\beta_\star^2(\mathbb{E}[\hat{\eta}] - \hat{\eta})^2. \tag{88}$$

Note that the condition is always satisfied in the case of the square loss (i.e., for $\delta \to +\infty \Leftrightarrow \bar{\chi} \to 1$).

**Consistency of the estimator.** The consistency of the estimator can be imposed by properly tuning $\lambda$, by requiring that $\lim_{\alpha \to +\infty}\varepsilon_{\text{est}} = 0$, i.e., $m = \beta_\star^2$ in this limit. In the same spirit as above, this implies a condition on $\lambda'$ given by

$$\lambda' = \hat{m} - \hat{v} = \mathbb{E}[\sigma^2\hat{\eta}\operatorname{erf}\bar{\chi}] - \mathbb{E}[\sigma^2\operatorname{erf}\bar{\chi}] \geq 0 \Rightarrow \mathbb{E}[\sigma^2(\hat{\eta} - 1)\operatorname{erf}\bar{\chi}] \geq 0 \tag{89}$$

to be computed with

$$\bar{\chi} \equiv \frac{\delta}{\sqrt{2\psi_{\text{est}}}}, \quad \psi_{\text{est}} = \hat{\sigma}_0^2 + \sigma_0^2\beta_\star^2(1 - \hat{\eta})^2. \tag{90}$$

When imposing the equality, the conditions above provide the values of $\delta$ (if any) for a consistent estimator if $\lambda = \Theta(1)$ in the $\alpha \to +\infty$ limit.

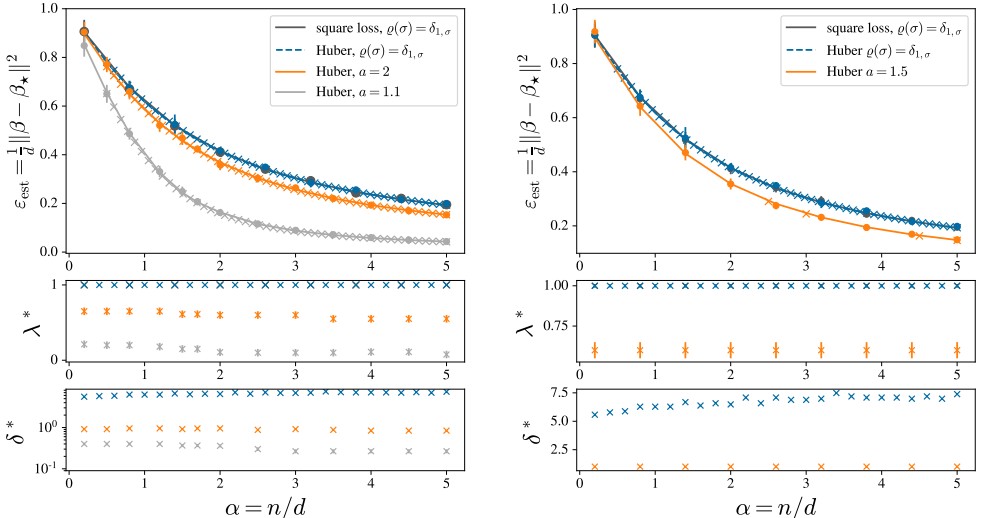

Figure 5: Gaussian covariates fully-contaminated by heavy-tailed noise with distribution $p_\eta(\eta) = \mathbb{E}_\sigma[\mathcal{N}(x; 0, \sigma^2)]$, where parameters of various $\varrho(\sigma)$ are varied: inverse-gamma (see Table. 1, **left**) and Pareto (**right**). (**Top**) Estimation error $\varepsilon_{\text{est}}$ as a function of the sample complexity $\alpha = {}^n/d$ for optimally regularised ridge regression (black), Huber with optimal location parameter and optimal regularisation (orange) and Bayes-optimal performance (crosses). (**Center.**) Value of the optimal regularisation parameter $\lambda^\star$ for the Huber loss. (**Bottom.**) Value of the optimal location parameter $\delta^\star$ for the Huber loss. Both optimal values are displayed by varying the scale parameter $a$ controlling the tails of the noise distribution. Dots indicate numerical simulations averaged over 20 seeds with $d = 10^3$.

## C  FURTHER NUMERICAL RESULTS

### C.1  FURTHER RESULTS FOR THE CASE OF HEAVY-TAILED NOISE

In this Appendix, we add some details about the case of $\epsilon$-contamination in the labels, as in Eq. 17, for different $\varrho_0$ generating the contaminating noise. Figure 5 compares the performance of various losses for different fully-contaminated ($\epsilon_{\text{n}} = 1$) label noise distributions, obtained picking for $\varrho_0(\sigma)$ taken to be inverse-Gamma as in Table 1 with $a = b + 1 > 1$ (left) or Pareto (right). In all cases, the chosen parametrisations enforce unit variance for the noise, $\mathbb{E}[\eta^2] = 1$. Taking the limit $a \to \infty$ in the inverse-Gamma or in the Pareto distributions results in recovering the Gaussian distribution for label noise. In our experiments, we observe the same phenomenology as in Fig. 2 (bottom) for all these densities which generate different noise label distributions. As long as the label noise variance is kept the same, optimally tuned regularised Huber loss performs the task better with heavier tails, in terms of the estimation Eq. 10. As in the cases discussed in the main text, optimally regularised Huber achieves Bayes-optimal performance.

## D  REBUTTAL PLOTS

This Appendix provides additional experiments requested by the reviewers.

### D.1  LEAST ABSOLUTE DEVIATION ESTIMATOR

We consider the LAD estimator corresponding to taking the $\ell_1$ loss $\ell_1(y, \eta) = |y - \eta|$ in Eq. 1b. Note this also corresponding to taking the location parameter $\delta$ of the Huber loss in Eq. 2 to zero.

Fig. 6 compares the optimally regularised LAD estimator with with the following estimators:

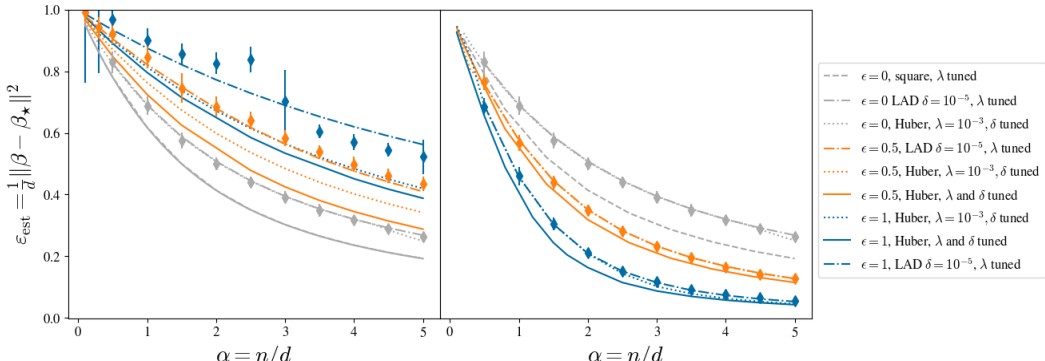

Figure 6: Response contamination including results for the LAD estimator, exactly on the same setting as Fig. 2. (**Left**) Infinite variance contamination. (**Right**) Finite variance contamination. Dots are experiments with dimensionality $d = 10^3$, averaged over 20 runs.

- Optimally regularised ridge regression.
- Unregularised Huber with optimally chosen location parameter $\delta$.
- Optimally regularised Huber with optimally chosen location parameter $\delta$

We consider three contamination cases $\epsilon \in \{0, 0.5, 1\}$, with both infinite and finite covariate variance. As it can be seen in Fig. 6, the LAD estimator performs similarly or worst to the other estimators in all cases considered.

## D.2 REVIEWER Fbsx'S ALGORITHM

In this Appendix, we numerically compare the performance the performance of Reviewer Fbsx's algorithm to standard ridge and Huber regression. The suggested algorithm consists of the following steps.

1. Given the training data $\mathcal{D} = \{(\boldsymbol{x}_i, y_i)_{i \in [n]} \in \mathbb{R}^{d+1}\}$, we uniformly sample a subset of size $^n/_2$, inducing a random partition of the training data $\mathcal{D} = \mathcal{D}_1 \cup \mathcal{D}_2$.
2. From subset $\mathcal{D}_1$ (wlog), we empirically estimate the median norm $\eta$ of the covariates $\boldsymbol{x}_i$.
3. From subset $\mathcal{D}_2$, we select only covariates which have norm smaller than $\eta$, i.e., we extract $\mathcal{D}_3 = \{(\boldsymbol{x}_i, y_i) \in \mathcal{D}_2 \colon \|\boldsymbol{x}\|_2 \leq \eta\}$.
4. Finally, we perform empirical risk minimisation Eq. 1b with either squared or Huber loss on $\mathcal{D}_3$.

Figure 7 numerically compares the performance of Reviewer Fbsx's algorithm to standard ridge and Huber regression on the full data set. We consider covariate contamination in the presence of heavy-tailed response noise. The suggested algorithm performs worse than all measures of performance for both the contaminated $\epsilon_c = 0.5$ (green) and fully-contaminated $\epsilon_c = 1$ (black) cases. This is true for both the square and Huber losses, using the optimal parameters obtained from our analysis on the full samples.

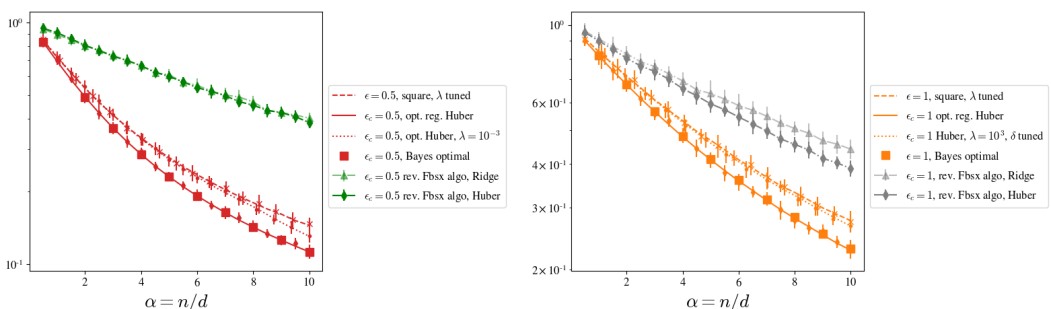

Figure 7: Comparison of the estimation error for vanilla empirical risk minimisation and Reviewer `Fbsx`'s thresholding algorithm. Covariate contamination with data whose variance is distributed as inverse-gamma variance with shape $a = 1.1$ and scale $b = 0.1$, in the presence of heavy-tailed response noise with variance distribution as inverse-gamma with parameters $a = 2, b = 1$.

