# OpenReview forum: "High-dimensional robust regression under heavy-tailed data: Asymptotics and Universality"
_ICLR.cc/2024/Conference — Submitted to ICLR 2024_

### Official Review · Reviewer_JXsv · 2023-10-30

**Soundness:** 3 good
**Presentation:** 3 good
**Contribution:** 3 good
**Rating:** 8
**Confidence:** 3

**Summary:**

This paper studies the high-dimensional asymptotics of the M-estimator trained over elliptical covariate and heavy-tailed noise distributions.  This is a general result where many distributions are covered. Theoretical guarantees are well developed, and generalizations to a richer family of models are also discussed.

**Strengths:**

This paper solves an open problem by depicting the asymptotics of the M-estimator over heavy-tailed contaminated models under the regime where $d/n\rightarrow \alpha>0.$ The paper rigorously presents the limiting behavior of the estimator and clearly discussed its generalizations.

**Weaknesses:**

I think there is no big weakness in this paper. My suggestion is to provide a more detailed explanation of the obtained results and compare them with previous benchmarks.

**Questions:**

This is a nice piece of work and I do not have many major questions. See my comments in the weakness section.

---

> ### Author Response · Authors · 2023-11-16
>
> We thank the reviewer for their positive evaluation of our paper. We remain attentive for further questions and suggestions during the discussion period.
>
> > *My suggestion is to provide a more detailed explanation of the obtained results and compare them with previous benchmarks.*
>
> We will provide a more detailed description of the results and the notation in a separate paragraph of the camera-ready version. Concerning benchmarks, we have implemented two suggestions by the other reviewers and compared them to our results, please refer to Appendix D.

---

### Official Review · Reviewer_sjuz · 2023-10-31

**Soundness:** 3 good
**Presentation:** 2 fair
**Contribution:** 2 fair
**Rating:** 6
**Confidence:** 3

**Summary:**

This paper studies the asymptotic performance of robust regression in the presence of heavy tailed data contaminated on both the covariates and the response.

The authors consider a model of the data that adds contamination on the covariates and the class of robust regression classifiers. They characterize, in the high-dimensional regime (where both sample size and dimension go to infinity at the same rate), the asymptotic alignment and the mean square error between the true signal and the predicted one as a function of the problem parameters. They also derive the asymptotic Bayesian optimal error. These results are both expressed as a function of low-dimensional quantities. Several intuitions are derived and experiments confirm the relevance of the approach.

**Strengths:**

1- The motivation to study the asymptotic performance of robust classifiers in the realistic regime, where the dimension is of the same order as the sample size, is interesting.

2- The results and intuitions derived seem interesting and new, and the theoretical and empirical predictions are in agreement with the experiments carried out.

3- The generality of the results, which can be applied to broader distributions such as Gaussian mixture model, is also an advantage, as is the independence of the formulae from any function used for robust regression.

**Weaknesses:**

1- The presentation could be improved, particularly the ratings. Given the wide range of notations, a section explaining the notational conventions would have been ideal, as would the rigour of defining all the notations and pointing them out. Although there is a section discussing the results, it would have been interesting to discuss results 2.4 and 2.6 a little better by showing particular cases where the solution is closed (such as ridge regression in a contaminated or uncontaminated case), and to derive the major changes in the theoretical formulae progressively.

2- It is a pity that the asymptotic performance depends on the true regression parameter, which limits its practical use (for example on real data) for optimizing the regularization parameters and loss in particular. The authors could have addressed the issue of estimation.

3- The results derived by the authors seem non-rigorous, although several attempts in the literature have made them fairly rigorous. It would be interesting to discuss the reasons why it is not possible to derive rigorous results and, above all, to explain why it is not possible to derive rigorous results. It would be interesting to discuss the reasons why it is not possible to derive rigorous results and above all to show to what extent the theoretical contribution of the paper is different and challenging from previous works.

**Questions:**

1- The paper mentions that the contamination model applies to both the covariates and the responses. However, in the definition of the model there is more mention of contamination on the covariates, which obviously indirectly induces contamination on the responses. But written this way, we would expect additional contamination on the regression noise.

2- Given the complexity of the expressions, it is important to simplify the notations as much as possible and to avoid defining unnecessary quantities and also to clearly define those which are used. Unless I'm mistaken, this is not always the case. For example in the result 2.4,  $\xi \sim \mathcal{N}(0, I_d)$ and $z$ seem to be defined but never used. Likewise, the integration variables $\eta$ and $\tau$ are not properly defined. The notation $\partial_w$ and $\partial_\mu$ are not clear since we do not know what $w$ and $\mu$ refer to.

3- The results depend on the real parameter $\beta_\star$, which limits the application of the bound on real data. Can the authors comment on when to estimate asymptotic performance to carry out, for example, model selection, ...


4- Section 4 could have been reorganized differently to provide more detailed explanation and experiments. In substance the result is, with the exception of one or two notations, the same as that of the main result. It is therefore in my opinion unnecessary to present it again. It can be put in an appendix by highlighting in the main paper what are the differences with the main result and deriving the main complementary intuitions.

5- The lack of rigor of the results still remains one of the weaknesses but it is very little discussed. What else prevents you from having rigorous results? What are the blocking passages?

After rebuttal, I read the answer of the authors. The authors promise to enhance the readability of the paper by providing notations section, by illustrating and discussion specific particular cases. I think this will allow to reach a wider audience. But I still think there are more to do in this sense. I will advise authors to progressively help the reader to grasp all the cumbersome notation by explaining before the theorem each main variables and giving intuitive explanation. How they behave in the asymptotic case (p fixed and n large)? What are the effects of the large dimensional corrections, and to particularly discuss notations that are specific to the heavy tailed analysis and that don't appear in the non heavy tailed analysis by raising some limiting cases. I think the problem of estimation of the signal to noise ratio limits the practical application of the paper and even though the authors claim it s a standard problem for regression problem, I think it is important to make the nice theoretical analysis practical even by providing some guidelines from the theory or from the analysis.

---

> ### Author Response · Authors · 2023-11-16
>
> We thank the reviewer for the useful comments and suggestions that helped to improve our work.
>
> > *1- The presentation could be improved, particularly the ratings. Given the wide range of notations, a section explaining the notational conventions would have been ideal, as would the rigour of defining all the notations and pointing them out. Although there is a section discussing the results, it would have been interesting to discuss results 2.4 and 2.6 a little better by showing particular cases where the solution is closed (such as ridge regression in a contaminated or uncontaminated case), and to derive the major changes in the theoretical formulae progressively.*
>
> We thank the referee for the suggestions: unfortunately, space constraints forced us to a compact presentation. Moreover, in the submitted version, we gave priority to presenting the most general result, which unfortunately comes with quite implicit expressions. Currently, closed form solutions for the square and Huber loss are given in Appendix B. In case of acceptance, by taking advantage of the additional provided space, we will add a dedicated paragraph for the discussion of the notation, as suggested by the referee: we also plan to move some of these results of Appendix B in the main text, providing more explicit and intuitive formulas for the most commonly used losses.
>
> > *2- It is a pity that the asymptotic performance depends on the true regression parameter, which limits its practical use (for example on real data) for optimizing the regularization parameters and loss in particular. The authors could have addressed the issue of estimation.*
>
> > *3- The results depend on the real parameter β⋆, which limits the application of the bound on real data. Can the authors comment on when to estimate asymptotic performance to carry out, for example, model selection, ...*
>
> We thank the referee for raising this point. We would like to stress that the asymptotic expression in Result 2.4 depends on the norm of $\beta_\star$ only, and not on the target parameter itself. This single scalar quantity, alongside the noise and covariates distribution widths, plays the role of a signal-to-noise ratio in the problem. The assumption on the existence of a true regression parameter is standard in the theoretical study of regression methods, and indeed is not too restrictive in the context of linear models, since one can learn, at best, a linear function of the covariates.
>
>
> > *3- The results derived by the authors seem non-rigorous, although several attempts in the literature have made them fairly rigorous. It would be interesting to discuss the reasons why it is not possible to derive rigorous results and, above all, to explain why it is not possible to derive rigorous results. It would be interesting to discuss the reasons why it is not possible to derive rigorous results and above all to show to what extent the theoretical contribution of the paper is different and challenging from previous works.*
>
> > *5- The lack of rigor of the results still remains one of the weaknesses but it is very little discussed. What else prevents you from having rigorous results? What are the blocking passages?*
>
> We expect that, as the referee correctly points out, our results can be put on rigorous ground using techniques from the high-dimensional statistics literature, e.g. Approximate Message methods or using Convex Gordon minmax inequalities. These techniques strongly rely on concentration inequalities that hold for Gaussian covariates, but not in the heavy-tailed case. Although we believe these techniques can be adapted using concentration inequalities for heavier-tailed random variables, the extension is non-trivial, and therefore we have opted to leave this for a future work devoted to this technical point. Note that for this reason we ran extensive finite-size simulations, all which show excellent agreement with our theoretical predictions as it can be seen in the Figures.
>
> > *1- The paper mentions that the contamination model applies to both the covariates and the responses. However, in the definition of the model there is more mention of contamination on the covariates, which obviously indirectly induces contamination on the responses. But written this way, we would expect additional contamination on the regression noise.*
>
> Indeed, our model allows for a contamination of both the noise and the covariates, and, for this reason, both are considered in the experiments. Following the reviewer’s suggestion, this will be made more explicit in the model definition and in the caption of the figures.

---

> ### Author Response · Authors · 2023-11-16
>
> > *2- Given the complexity of the expressions, it is important to simplify the notations as much as possible and to avoid defining unnecessary quantities and also to clearly define those which are used. Unless I'm mistaken, this is not always the case. For example in the result 2.4,  $\xi\sim\mathcal{N}(0,Id)$ and $z$ seem to be defined but never used.*
>
> We thank the referee for pointing out these sources of confusion. We have clarified them in the updated version. As anticipated in a previous answer, we will be including a dedicated paragraph to clarify the notation adopted in the paper in the camera-ready version with additional space.
>
> > *4- Section 4 could have been reorganized differently to provide more detailed explanation and experiments. In substance the result is, with the exception of one or two notations, the same as that of the main result. It is therefore in my opinion unnecessary to present it again. It can be put in an appendix by highlighting in the main paper what are the differences with the main result and deriving the main complementary intuitions.*
>
> The result presented in Section 4 is considerably more general than the one in Section 2, covering for instance multi-modal elliptical distributions and generic convex regularisation (e.g. LASSO). However, since our focus was on robust regression, we have decided to present it separately to avoid having even more cumbersome equations. Result 4.3 also allows us to extend known universality results for Gaussian covariates to the elliptical case, which are discussed in Appendix B. In the revised version, we will be moving these results to the main in order to provide better context for Section 4.

---

> ### Comment · Reviewer_sjuz · 2023-11-23
> **Answer to rebuttal**
>
> I would like to thank the authors for the time they took to answer the questions mentioned.
> I think that a notation section would greatly improve the readability of the article as well as the discussion of specific cases of the different formulas. As mentioned by several other reviewers, the results need to be discussed and explained in depth to allow readers to grasp the insights.
>
> Although I understand that performance depends on the norm of the true regression vector rather than the vector, estimating it is still problematic. What happens if we replace the norm of the true regression vector with the norm of the empirical vector? Is it biased? Are there any corrections to be made? Indeed, we expect performance to depend on the signal-to-noise ratio, but estimating it remains a major problem for practical analysis. The noise level also needs to be estimated.
>
> Although with all these residual problems I increase my score to 6 and explain the main reasons in the main review in the section Questions.

---

### Official Review · Reviewer_Fbsx · 2023-10-31

**Soundness:** 3 good
**Presentation:** 2 fair
**Contribution:** 2 fair
**Rating:** 5
**Confidence:** 3

**Summary:**

The paper studies linear regression with elliptically distributed designs (covariates). They show asymptotic behaviour of general $\ell_2$ penalised M-estimators (e.g. squared loss or Huber loss), as well as the asymptotic behaviour of the Bayes-optimal estimator.

**Strengths:**

The results are new and non-trivial, and the problem is interesting.

**Weaknesses:**

While I find linear regression with heavy-tailed designs (e.g. elliptically distributed) and heavy-tailed noise (e.g. Cauchy) interesting, there are some things in the paper that I didn't like. See the questions section for more details. Currently I don't think the paper is above the acceptance threshold, so I vote for 5 (marginally below the acceptance threshold).

**Questions:**

1) I couldn't fully understand Result 2.4, I think this is too abstract. Result 2.5. is clear, but I couldn't understand what you get if you use Huber loss for $\eta_i$ with infinite variance. Could you say what happens in this case? (You can use the same distribution of $\eta_i$ that you used for experiments, or any other meaningful distribution with infinite variance, I just want to understand the result and compare it with 2.5). I'm also fine with a simplified answer that is not very precise and ignores a factor bounded by a universal constant.

I also suggest you to explicitly write a separate general result for unbounded variance noise (and Huber loss) somewhere in the main part of the paper (again, maybe simplifying the formulas and making the bounds worse by a constant factor). To do this, you should fix some general property of $\eta$. As you observed at the end of page 19, one may assume wlog that the symmetry assumption (Assumption B.2) is satisfied in this case. In addition, I think you need to fix the scale of the noise. For this you may use some assumptions from prior works, e.g. the classical assumption from [Pol91] that $\eta$ has some positive density at zero. Or perhaps you already used something similar somewhere in the paper but I missed it?

2) I think some discussion related to Result 2.5 and the state of the art would be good to add. In particular, if you normalize $X$ so that $X^\top X = I$ and then do ridge regression, then the parameter error is well-known, and then you can go back to the old basis by dividing the error by the smallest singular value of $X^\top X$. Can you compare this simple approach with your result, is your result strictly better?

3) Does the following simple approach provide results comparable to yours: First we use half of the covariates to learn the median norm of $x_i$, and then we only take samples from the second half that are less than this median norm. Then the covariates should be not so bad and it might be possible to use known results for squared or Huber loss. Is it strictly worse than your result? It would be good to add a comparison to the paper.

4) What do you mean by "non-rigorous" when you discuss Results 2.4 and 2.6 on page 5?

Less important:

5) For the unbounded variance noise (e.g. the one that you consider in the experiments, or, again, any other meaningful noise), does the ($\ell_2$-penalized) LAD estimator have similar guarantees as Huber loss, or are they asymptotically different here?

[Pol91] David Pollard, Asymptotics for least absolute deviation regression estimators, Econometric Theory 1991

---

> ### Author Response · Authors · 2023-11-16
>
> We thank the reviewer for the useful comments and suggestions that helped to improve our work.
>
> > *I couldn't fully understand Result 2.4, I think this is too abstract.*
>
> We acknowledge that the formulas in Result 2.4 can be hard to parse. However, on a high-level, it conveys a simple message: any statistics for the model defined in Assumptions 1-3, including for example the test error or mse on the parameters, can be exactly computed by solving a set of low-dimensional equations (12a) in the high-dimensional limit. The equations themselves are cumbersome due to the generality of the result (arbitrary convex loss, general label likelihood and covariate variance distribution, etc.), but can be efficiently implemented in a computer and solved to arbitrary precision since they are dimension-free.
>
> > *Result 2.5. is clear, but I couldn't understand what you get if you use Huber loss for with infinite variance. Could you say what happens in this case? (You can use the same distribution of  that you used for experiments, or any other meaningful distribution with infinite variance, I just want to understand the result and compare it with 2.5). I'm also fine with a simplified answer that is not very precise and ignores a factor bounded by a universal constant.*
>
> Our result in Fig. 4 (right) shows that under the same assumptions of Result 2.5, Huber achieves the same rates as the square loss, for both finite and infinite variance. Due to the particularly simple structure of the proximal operator for the square loss, eq. (12a) can be reduced to a single equation for the mse, which allows for the rates to be analytically extracted (Result 2.5). Unfortunately, this is not the case for the Huber loss, for which the equations remain cumbersome and the rates have to be extracted by solving the equations numerically.
>
> > *I also suggest you to explicitly write a separate general result for unbounded variance noise (and Huber loss) somewhere in the main part of the paper (again, maybe simplifying the formulas and making the bounds worse by a constant factor). To do this, you should fix some general property of η. As you observed at the end of page 19, one may assume wlog that the symmetry assumption (Assumption B.2) is satisfied in this case. In addition, I think you need to fix the scale of the noise. For this you may use some assumptions from prior works, e.g. the classical assumption from [Pol91] that η has some positive density at zero. Or perhaps you already used something similar somewhere in the paper but I missed it?*
>
> We thank the reviewer for this suggestion. We will be adding a section discussing the infinite noise variance case separately in the revised version.
>
> > *I think some discussion related to Result 2.5 and the state of the art would be good to add. In particular, if you normalize $X$ so that $X^{\top}X=I$ and then do ridge regression, then the parameter error is well-known, and then you can go back to the old basis by dividing the error by the smallest singular value of $X^{\top}X$. Can you compare this simple approach with your result, is your result strictly better?*
>
> Under our data model, the case $X^{\top}X=I$ corresponds exactly to the abundant data regime with Gaussian tail, and therefore can be recovered as the $\alpha = n/d \to\infty$  of our results. The same is true for the heavy-tailed cases, by changing the identity to the variance of the data. Therefore, our approach is considerably more general: it covers not only the classical statistical regime $n\gg d$ but also the the high-dimensional regime where $d>n$ (where the sample covariance $X^{\top}X$ is very far from the population $I$), which has drawn significant interest motivated by the study of overparametrisation in machine learning.
>
> > *Does the following simple approach provide results comparable to yours: First we use half of the covariates to learn the median norm of xi, and then we only take samples from the second half that are less than this median norm. Then the covariates should be not so bad and it might be possible to use known results for squared or Huber loss. Is it strictly worse than your result? It would be good to add a comparison to the paper.*
>
> One take-away from our results is that when properly regularised by cross-validation, minimising the Huber loss achieves close to optimal statistical performance, therefore doing better (or equal) than any other estimator having access only to a finite sample of training data. For an explicit comparison, we have run simulations for the suggested algorithm in the case of covariate contamination (both finite and infinite variance) and heavy-tailed noise. Figure 7 in Appendix D of the updated manuscript compares the suggested algorithm against the vanilla ERM ridge and Huber regression, showing it underperforms in all cases investigated.

---

> ### Author Response · Authors · 2023-11-16
>
> > *What do you mean by "non-rigorous" when you discuss Results 2.4 and 2.6 on page 5?*
>
> Results 2.4 & 2.6 are derived using the replica method, a heuristic tool inspired by statistical physics.  Note that similar expressions derived with this method have been made rigorous for several problems in high-dimensional statistics and machine learning, using techniques such as Gordon’s minmax inequalities or Approximate Message Passing, see refs. [1,2,3]. However, generalising these proof schemes to heavy-tailed covariates is a non-trivial extension, since they rely on concentration inequalities which don’t hold in this case. For this reason, we have added finite-size simulations to all our Figures, beside running extensive additional checks. All finite-size simulations show an excellent agreement with the theoretical formulas, suggesting this is a technical, rather than fundamental, extension.
>
> [1] Deng et al. https://arxiv.org/abs/1911.05822 (2019)
>
> [2] Mignacco et al., https://arxiv.org/abs/2002.11544 (2020)
>
> [3] Gerbelot and Berthier, https://arxiv.org/abs/2109.11905 (2021)
>
> > *For the unbounded variance noise (e.g. the one that you consider in the experiments, or, again, any other meaningful noise), does the (l2 penalized) LAD estimator have similar guarantees as Huber loss, or are they asymptotically different here?*
>
> The LAD estimator is covered by our results, since it corresponds exactly to the $\delta = 0$ case of the Huber loss. In particular, note that we observe an interesting “phase transition” as a function of the sample complexity between a regime where the LAD estimator is sub-optimal and a regime where it is optimal, which is discussed in Figure 1. Investigating better this transition is an interesting avenue for future work.
>
> Nevertheless, following the reviewers request we have run some theoretical curves and the corresponding numerical simulations for optimally regularised LAD in the same setting as Figure 2 from the manuscript. This is shown in Figure 6 in the new Appendix D of the updated manuscript.

---

> > ### Comment · Reviewer_Fbsx · 2023-11-23
> >
> > Dear Authors,
> >
> > Thank you very much for the response and the clarification! I apologize for such a late reply, I really could not reply earlier...
> >
> > Perhaps I was not very clear about $X^\top X = I$. I just meant that one can do a different approach: take some (heavy-tailed) $X$ and *then* normalize it so that $X^\top X = I$ , and then solve ridge regression. This is a simple approach and perhaps it is not the best, but I think it is good to compare it with your results.
> >
> > Regarding other things, I don't have further questions. Currently I don't increase the score, but I will take into account your response during the Reviewer/AC Discussion.

---

### Author Response · Authors · 2023-11-22
**Summary of changes**

Dear reviewers,

We would like to thank you again for the useful feedback that helped improving our manuscript.

As the discussion period comes to an end, we would like to draw your attention to the revised manuscript we have uploaded. We have added an Appendix D "*Rebuttal plots*" with additional experiments and incorporated your suggestions within the page constraints.

In particular:
- As requested by Reviewer *Fbsx*, we have added theoretical curves for the LAD estimator for both the finite and infinite variance case, under different contamination levels.
- We have numerically implemented the algorithm suggested by Reviewer *Fbsx*, showing it underperforms the other methods studied in the paper for both finite and infinite variance and different contamination levels.
- We have implemented all the small changes suggested by the three reviewers.

Moreover, we plan to add the following with the extra space provided in the camera-ready version:

- Add a separate section for discussing the infinite variance case, as suggested by Reviewer *Fbsx*.
- Add a notation section, as suggested by Reviewer *sjuz*.
- Add an additional paragraph discussing the Result 2.4 intuitively, as suggested by both Reviewer *sjuz* and Reviewer *JXsv*.

We remain attentive if you have any additional suggestions and/or questions.

The authors.

---

### Meta-Review · Area_Chair_mNVi · 2023-12-22

**Metareview:**

This is a paper in mathematical statistics, which aims to characterize M-estimators and the Bayes estimator in a linear model in an asymptotic regime where the number of features and number of data are comparable. Through the analysis of the mean squared error of the M-estimator, the authors show that Huber loss may be suboptimal in the high dimensional regime under a heavy-tailed response contamination. (These results are in contrast to the optimality of Huber loss in standard asymptotic regimes that are low dimensional.) The authors propose new methodology to achieve optimality.
This is not yet a result in machine learning, as the paper should contain a discussion of the results and implications for machine learning. Since the paper assumes a linear model, I do not see much impact for machine learning. Traditional machine learning theory would not assume a model for the data, but instead characterize excess risk in a distribution free setting, maybe aiming to achieve adaptivity to some parameter characterizing easiness.

Even so, the reviewers found the presentation of the theoretical results challenging. (It’s written for a stats audience not a ML one). The reviewers would like the notation to be simplified, multi-equation lines to be broken up with explanations, etc. Including a higher level proof overview of the main results would also be a good addition, further improving readability.

**Justification For Why Not Higher Score:**

The paper is not an ML theory paper, but a result in mathematical statistics.

**Justification For Why Not Lower Score:**

Overall, an interesting paper with some fundamental results of M- and Bayes-estimators in a high dimensional regime and contaminated data. Will likely be of interest to the small subset of ML researchers who work also in mathematical statistics.

---

### Decision · Program_Chairs · 2024-01-16

Reject